# Highly parallel and ultra-low-power probabilistic reasoning with programmable gaussian-like memory transistors

Changhyeon Lee[1,6], Leila Rahimifard[2,6], Junhwan Choi [3,6], Jeong-ik Park[1], Chungryeol Lee[1], Divake Kumar[2], Priyesh Shukla[2], Seung Min Lee[1], Amit Ranjan Trivedi [2] ✉, Hocheon Yoo [4] ✉ & Sung Gap Im [1,5] ✉

Probabilistic inference in data-driven models is promising for predicting outputs and associated confidence levels, alleviating risks arising from overconfidence. However, implementing complex computations with minimal devices still remains challenging. Here, utilizing a heterojunction of p- and n-type semiconductors coupled with separate floating-gate configuration, a Gaussian-like memory transistor is proposed, where a programmable Gaussian-like current-voltage response is achieved within a single device. A separate floating-gate structure allows for exquisite control of the Gaussian-like current output to a significant extent through simple programming, with an over 10000 s retention performance and mechanical flexibility. This enables physical evaluation of complex distribution functions with the simplified circuit design and higher parallelism. Successful implementation for localization and obstacle avoidance tasks is demonstrated using Gaussian-like curves produced from Gaussian-like memory transistor. With its ultralow-power consumption, simplified design, and programmable Gaussian-like outputs, our 3-terminal Gaussian-like memory transistor holds potential as a hardware platform for probabilistic inference computing.

With the remarkable strides of AlphaGo, the integration of deep learning into artificial intelligence (AI) has assumed a pivotal role across diverse industrial domains, encompassing autonomous driving[1–3], image recognition[4–6], and translation[7,8]. However, the intricate interplay of environmental factors such as variability, occlusion, and fluctuations in lighting, coupled with sensor constraints like noise, limited resolution, and range, collectively employ a considerable multitude of uncertainties in perception[9–11]. Regrettably, conventional AI systems, which generally disregard such kinds of uncertainty, often yield inaccurate predictions and compromise the dependability of inferences from deep learning models. To combat this overconfidence quandary, a critical imperative emerges: adopting a calculation methodology that can faithfully embrace uncertainty[12,13]. Probabilistic inference procedures, rooted in probability theory and statistical techniques, rise to the challenge, skillfully navigating the complexities of diverse uncertainties in perception and learning. By adeptly representing and reasoning unfamiliar predictive models or ambiguous data patterns that breed uncertainty, probabilistic inference models empower decision-makers to exercise heightened discernment and adaptability in the face of evolving circumstances[14–16].

[1]Department of Chemical and Biomolecular Engineering, Korea Advanced Institute of Science and Technology (KAIST), 291 Daehak-ro, Yuseong-gu, Daejeon 34141, Korea. [2]Department of Electrical and Computer Engineering, University of Illinois at Chicago, Chicago, IL 60607, USA. [3]Department of Chemical Engineering, Dankook University, 152 Jukjeon-ro, Suji-gu, Yongin, Gyeonggi-do 16890, Korea. [4]Department of Electronic Engineering, Gachon University, 1342 Seongnam-daero, Sujeong-gu, Seongnam, Gyeonggi-do 13120, Korea. [5]KAIST Institute for NanoCentury (KINC), Korea Advanced Institute of Science and Technology (KAIST), 291 Daehak-ro, Yuseong-gu, Daejeon 34141, Korea. [6]These authors contributed equally: Changhyeon Lee, Leila Rahimifard, Junhwan Choi. ✉e-mail: amitrt@uic.edu; hyoo@gachon.ac.kr; sgim@kaist.ac.kr

However, the decision-making with probabilistic models also necessitates substantial computational resources[17–19]. Gaussian distribution and its mixtures are typically employed in probabilistic inference procedures[20]. The central limit theorem states that the sum of many independent and identifiably distributed random variables, regardless of their underlying distribution, can be approximately normally distributed[21]. By combining multiple Gaussian functions through a mixture, multi-modal statistical data can be efficiently modeled. Nevertheless, evaluating functions of Gaussian mixture model (GMM) using a digital data-path often results in excessive computational workload[22,23]. Notably, the computation of a GMM's output entails a multitude of multiplications, additions, and look-up operations. Such workload in computation grows with the signal dimension and the complexity of the model, determined by the number of mixture functions in the GMM[24]. Given the importance and versatility of GMM, previous studies have investigated GMM computation hardware intensively utilizing dynamic random-access memory (DRAM)[25], field-programmable gate array (FPGA)[26–28], or analog circuits. However, usually a large number of transistors was required, which inevitably resulted in a high level of complexity[29]. Consequently, without innovative technologies that can dramatically reduce resource demands when dealing with high-dimensional complex distribution functions like GMMs, the advantages of probabilistic inference procedures remain out of reach for numerous applications.

Pursuing a probabilistic inference, significant strides have been made in recent years to develop devices capable of physically emulating the characteristics of a Gaussian function[30–33]. This physics-based computing approach, using the Gaussian devices, holds the potential for a revolutionary enhancement in energy efficiency, thereby enabling efficient operation of GMMs with highly constrained resources. For instance, analog circuits based on silicon-based complementary metal–oxide–semiconductor (CMOS) technology have been used to emulate Gaussian functions. However, in conventional Si metal-oxide-semiconductor field-effect transistor (MOSFET)-based circuits, at least four transistors are required to represent a Gaussian distribution[34,35]. Michail et al. presented a GMM-based classifier by connecting NMOS and PMOS bump circuits with a winner-takes-all (WTA) circuit[36]. Another study, Vassilis et al. proposed low-power bell-shaped analog classifiers (CLFs) by implementing GMM (machine learning) circuits with fewer number of transistors and three input values, and tested on real-world biomedical classification problems[37]. Nevertheless, to tune the characteristics of the distribution function, the number of transistors increases often to more than 10, rendering the circuit quite complex and inevitably increasing power consumption[31,32,38,39].

Here, we show a device, called Gaussian-like memory transistor (GMT), yielding a Gaussian-like current ($I$)−voltage ($V$) relationship in a single heterojunction transistor device. Moreover, each semiconductor has a separate floating gate (FG), allowing independent adjustment of their channel conductivity, thereby enabling systematic control of Gaussian-like expression through the FG. The GMT demonstrates controllable Gaussian-like $I - V$ characteristics based on programming states of the semiconductors, exhibiting over 10000 s retention, operational stability against repeated cycles. Leveraging GMT's Gaussian-like behavior and controllability, we pioneer a component architecture that allows for the implementation of inference capabilities for probabilistic robotics applications through a remarkably simple design. The GMT devices possess the remarkable capacity to adjust Gaussian-like shapes extensively, all the while maintaining an impressively low power consumption profile. This attribute empowers us to offer practical calculations that effectively mitigate the challenge of overconfidence.

## Results

### Design of the Gaussian-like memory transistor

A schematic diagram of our approach is shown in Fig. 1a–c. We adopt a Bayesian filtering approach for different probabilistic inference tasks, and construct the operational framework using GMM-based probability distribution functions within GMT arrays[40–43]. By modeling the likelihood function as a mixture of Gaussian functions, the posterior weight density storage can be computed. Although leveraging distribution functions enables energy-efficient, low-latency, and on-board processing, minimizing evaluation workload, representing a single probability distribution function necessitates numerous devices to perform multiplication and accumulation operations (Fig. 1d, e)[19,44–47]. To effectively handle diverse situations through precise control over the output format of Gaussian distributions, this work proposes a simple 3-terminal GMT device capable of representing precisely tunable Gaussian-like distributions (Fig. 1e). The GMT utilizes an organic semiconductor heterojunction, where a p-type semiconductor is connected to an n-type counterpart at the midpoint of the channel layer, inducing anti-ambipolar characteristics shaped by each semiconductor's subthreshold region. To achieve Gaussian distribution-like symmetric outputs, N,N′-ditridecylperylene-3,4,9,10-tetra-carboxylic diimide (PTCDI-C13) and pentacene were utilized as n-type and p-type semiconductors, respectively, owing to their comparable charge transport performance[48,49] (Supplementary Fig. 1).

For systematic controllability of Gaussian-like output characteristics by independent modulation of p- and n-type channel conductance, separate FG structures were introduced beneath both p- and n-type semiconductors. Gold nanoparticles (AuNPs) served as separate FGs due to their high charge-loss tolerance resulting from discrete spatial distribution[50–52] (Supplementary Fig. 2). The independent control of channel conductivity of each semiconductor through FGs allows for versatile and precise control over the characteristics of the Gaussian-like $I − V$ characteristics, enabling efficient modeling of complex probability distribution functions as well as representing the target distribution by connecting GMT arrays (Fig. 1f, g). To achieve memory performance, careful consideration of both the floating gate layer and several other insulating layers are important. The blocking dielectric layer (BDL) must possess exceptional insulating characteristics to impede charge leakage and current flow to the gate electrode. Additionally, the tunneling dielectric layer (TDL) should not only facilitate tunneling during programming voltage application but also ensure the non-leakage of stored charge. Therefore, the TDL requires appropriate insulating properties, especially under low-thickness conditions. By strategically placing the vacuum-deposited polymer dielectric films with different electrical properties at appropriate locations within the device[50,53–56]. A significant reduction in operating voltage and programming/erasing voltage of the GMT device was achieved, which is crucial in the realization of low-power, energy-efficient GMT arrays (Supplementary Fig. 3).

The high-resolution transmission electron microscope (HRTEM) images, optical microscope (OM) images and corresponding energy dispersive spectroscopy (EDS) mapping clearly confirm that the flash memory structure was successfully implemented (Fig. 2a–c and Supplementary Fig. 4). As shown in the optical microscope (OM) image, the GMT was operated by connecting pentacene and PTCDI-C13 into drain ($D$) and source ($S$) electrodes, respectively. As shown in Fig. 2d, a Gaussian distribution-like current output was indeed implemented in the GMT device, regardless of different drain voltages ($V_D$). The operating voltage of the GMT devices was reduced to be less than 3 V with further scaling down the thickness of dielectric layers (Supplementary Fig. 17). With the variation in $V_D$, only off voltage ($V_{off}$) was affected, while on voltage ($V_{on}$) remained constant, causing the variation of $V_{off}$ with the change in $V_D$[57,58] (Supplementary Fig. 6). The Gaussian distribution-like $I − V$ transfer characteristics were modeled through the following equations below[59]:

$$I_D = A \exp\left[-\frac{(V_{BG} - \mu)^2}{2\sigma^2}\right]; \qquad (1)$$

where the $A$, $\mu$, and $\sigma$ represent the amplitude, mean, and standard deviation of the gaussian function, respectively. A strong correlation between the experimentally measured data and fitted curves was observed (Supplementary Fig. 6). The operational stability of our GMT device was also monitored where Gaussian distribution-like transfer curve was fully preserved in the repetitive operation (Fig. 2e). Over 100 consecutive operations, only marginal changes in $\mu$ (less than 100 mV) and $\sigma$ (less than 30 mV) was observed (Fig. 2f). Furthermore, uniform operation was demonstrated across 8 devices, securing the potential for large-scale circuits (Supplementary Fig. 7).

### Individual programming of channel conductance

For the hardware implementation of diverse probabilistic model within a single device, it is highly desirable to retain precise control of the Gaussian distribution characteristics[38,60]. To validate the controllability of the channel conductivity through the separate FGs, we analyzed the change in $I-V$ characteristics by the programming operation, as shown in Fig. 3a. It should be noted that we applied reverse bias to $D$ (or $S$) electrode for programming p-type (or n-type) semiconductor, instead of applying $V_G$ directly into the global gate electrode[53,55,61]. We refer to this inverse bias for p- and n-type semiconductors as programming voltage of p-type ($V_{prg,P}$) and n-type semiconductor ($V_{prg,N}$), respectively.

The changes in transfer curves depending on programming operation of n-type depicted in Fig. 3b. These programming operations were investigated by using an incremental step pulse programming (ISPP) technique[55,62], with a common pulse width set to 1 s. The shape of the transfer curve of the GMT device is determined by the conductivity of the p- and n-channel. In other words, the current switching behavior of the GMT device is closely dependent on the $V_T$ of p-channel ($V_{T,P}$) and n-channel ($V_{T,N}$). When a programming voltage is applied, electrons (or holes) trapping occurs through tunneling, similar to conventional floating gate memory[53,55,61]. When the electrons are stored in the FG, they induce an additional negative voltage, causing the $V_T$ shift toward positive direction. Analogously, stored holes force the $V_T$ shift toward negative direction. The $V_{T,P}$ and $V_{T,N}$ were gradually changed along with the applied $V_{prg,P}$ and $V_{prg,N}$, respectively. Initially, the $V_T$ values of each channel layer were measured to be $V_{T,N} = 2.01$ V and $V_{T,P} = 4.68$ V. As shown in Fig. 3c, with the increasing absolute value of $V_{prg,N}$, the amount of holes (or electrons) in the n-type FG increased. Therefore, the change of $V_{T,N}$ ($\Delta V_{T,N}$) reached +1.20 V and −1.35 V at the $V_{prg,N} = +22$ V and $V_{prg,N} = -22$ V, respectively.

Similarly, the transfer curve changes depending on programming operation p-type semiconductors depicted in Fig. 3d, and by p-type programming operation, $V_{T,P}$ also experienced a shift; the change of

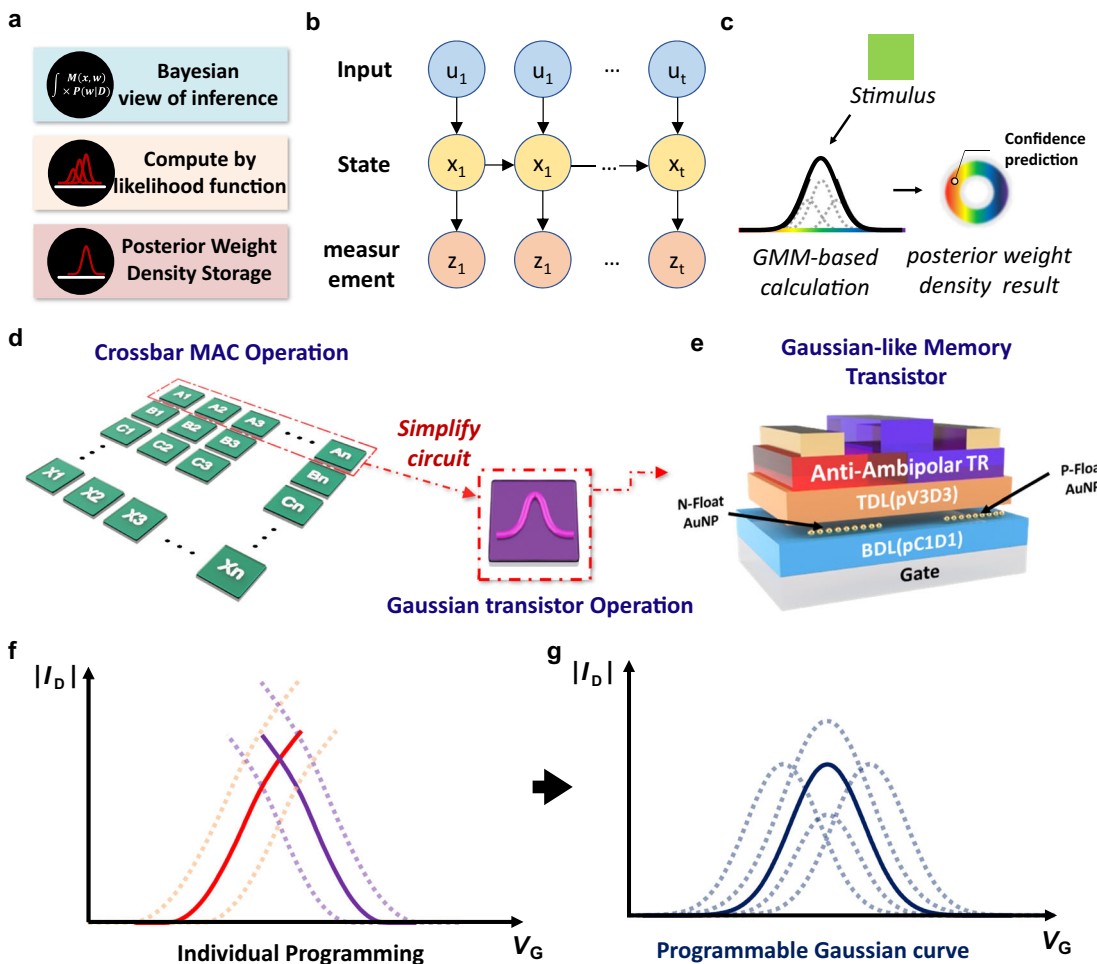

**Fig. 1 | Design of Gaussian-like memory transistor. a** Resurrection of three Bayesian filtering approaches for various probabilistic inference tasks. **b** Schematic representation of Bayesian filtering approach that comprises of an input and state and measurement. **c** Schematic representation of probabilistic inference that comprises a probability distribution function layer for Gaussian mixture model (GMM) calculation input stimulus to the output value. **d** A schematic concept of the probabilistic reasoning using a Gaussian-like output transistor compared to the multiply and accumulation (MAC) operation. **e** a schematic illustration of a Gaussian-like memory transistor (GMT) device structure. **f, g** The schematic illustrations of the p- (purple) and n-type (red) transfer curve (absolute value of drain current ($|I_D|$) versus gate voltage ($V_G$)) shift along with the memory programming (**f**) and corresponding transfer characteristic (purple) of the GMT device (**g**).

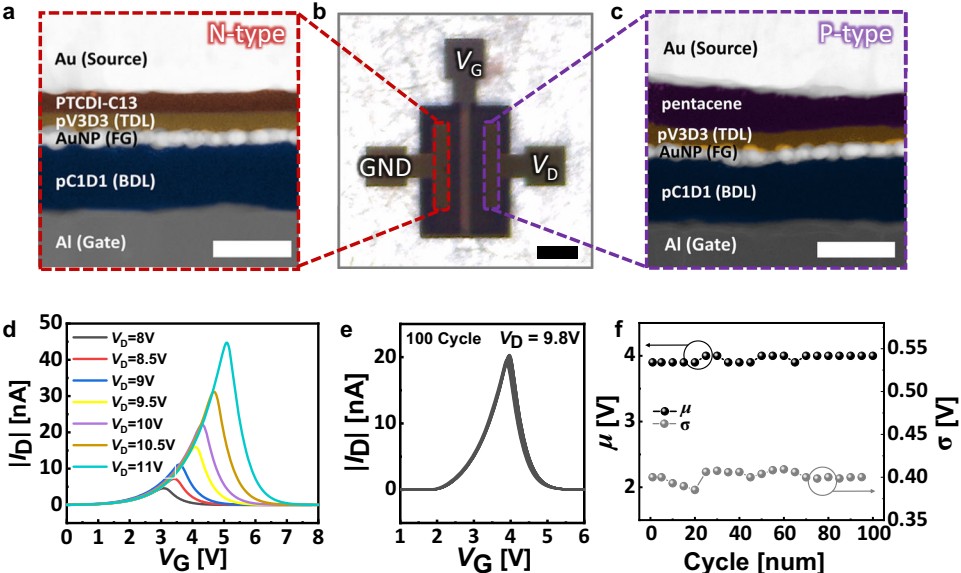

**Fig. 2 | The structure and basic operation of GMT device. a** The cross-sectional high-resolution transmission electron microscope (HRTEM) images along with n-type semiconductor region (scale bar = 50 nm). **b** The optical microscope (OM) image (scale bar = 500 μm) (**b**). **c** The HRTEM images along with p-type semi-conductor region. The false-color modification was attempted to distinguish each layer in HRTEM images (scale bar = 50 nm). **d** The transfer characteristics (absolute value of drain current ($|I_D|$) versus gate voltage ($V_G$)) of Gaussian-like memory transistor (GMT) device with different drain voltage ($V_D$). **e, f** The transfer curve of 100 consecutive sweeps (**e**) and mean ($\mu$) and standard deviation ($\sigma$) values (**f**) of the GMT device at drain voltage ($V_D$) = 9.8 V.

$V_{T,P}$ ($\Delta V_{T,P}$) reached −1.09 V and +2.12 V at the each point of $V_{prg,P}$ = −20 V and $V_{prg,P}$ = +24 V (Fig. 3e). These observations confirm that the programming on each FG beneath the corresponding channel layer can efficiently suppress unintended programming operation of FG on the other side, and enables the independent control of the n- and p-channels through the programming operation. Thanks to these advantageous attributes, while one side's threshold voltage ($V_T$) shifts, the other remains unchanged and consistently maintained.

**Gaussian factor control**
Beyond the accomplished control of Gaussian-like distribution through individual FG programming, the concurrent programming through the desirable ways allows for the meticulous manipulation of $\mu$ or $\sigma$, while ensuring minimal interference with other parameters (Fig. 4a, b). For probabilistic inference operations applying Bayesian filtering, it is essential to adjust $\mu$ and $\sigma$ in a way that avoids interference between them. Therefore, the programming process is designed to minimize variations in $\sigma$ when $\mu$ changes, and vice versa, ensuring that changes in $\sigma$ result in minimal alterations to $\mu$. Based on the programming method suggested above, we devise two distinctive programming methods to control the Gaussian-like distribution of our GMT device. Case 1 is the injected charges have the same polarity and quantity, an equal amount of $V_T$ shift arises in each channel layer ($\Delta V_{T,P} = \Delta V_{T,N}$). This leads to a parallel shift in the transfer curve ($\mu$ regulation) without perturbing $\sigma$, and the direction of the $V_T$ shift corresponds to the polarity of charges injected through FGs (Fig. 4a). Case 2 is when the injected charges have opposite polarity and the same quantity, and an opposite direction of $V_T$ shift occurs with the same magnitude ($\Delta V_{T,P} = -\Delta V_{T,N}$), This results in a shape change of the transfer curve, leading to $\sigma$ regulation while retaining the constant $\mu$ (Fig. 4b).

Figure 4c, d demonstrates the change in the Gaussian distribution-like transfer curve when the injected charge carriers have the same polarity and quantity ($\Delta V_{T,P} = \Delta V_{T,N}$, Case 1). When holes were trapped on both n- and p-type FGs, the $V_T$ of both channel layers shifted in the negative direction to the same extent, resulting in a parallel movement of the Gaussian-like curve towards the negative direction (Fig. 4c).

Likewise, when electrons were trapped in both FGs, the $V_T$ shifted towards the positive $V_G$ direction for both channel layers, leading to a parallel movement of the Gaussian-like curve towards the positive direction without changing the Gaussian-like curve shape (Fig. 4d). As shown in Fig. 4e, due to the $V_T$ shift induced by the amount of trapped charges, a near-linear relationship between $\mu$ and the programming voltage was achieved[61]. The initial $\mu$ value was observed to be 4 V and it gradually shifted after programming. The $\mu$ value reached 2.1 V with a negative movement ($V_{prg,P}$ = −25 V and $V_{prg,N}$ = −24 V) and reached 6.4 V with a positive movement ($V_{prg,P}$ = +23 V and $V_{prg,N}$ = +22 V), which is substantial variation considering the width value of 1.92 V set as $6\sigma$ following the $3\sigma$ rule in Supplementary Note 4. In the parallel shift of the transfer curve, only practically negligible variation was detected in $\sigma$ (Fig. 4e).

Under the condition where the injected charge carriers have different polarity ($\Delta V_{T,P} = -\Delta V_{T,N}$, Case 2), the $\sigma$ regulation could be accomplished (Fig. 4f, g). When holes were trapped on the n-type FG and electrons were trapped on the p-type FG, $V_{T,N}$ and $V_{T,P}$ moved in opposite directions, leading to the widening of the Gaussian distribution-like output (Fig. 4f). Conversely, when the opposite charges are trapped, $V_{T,N}$ and $V_{T,P}$ moved closer to each other, resulting in the narrowing the distribution (Fig. 4g). The changes in the $\sigma$ value decreased to 0.23 V at the $V_{prg,P}$ = −22 V and $V_{prg,N}$ = +20.5 V from the initial value of 0.32 V, and increased to 0.54 V at the $V_{prg,P}$ = +21.5 V and $V_{prg,N}$ = −21 V as shown in Fig. 4h. While $\sigma$ undergoes systematic change, the $\mu$ value remain virtually constant (Fig. 4h). Detailed programming conditions of $\mu$ and $\sigma$ regulations are described in Supplementary Fig. 13. These results clearly demonstrate that the adjustment of Gaussian distribution-like $I − V$ characteristics including $\mu$ and $\sigma$ are successfully achieved by simple programming operation by a single 3-terminal heterojunction device. These results demonstrate that GMT device characteristics can be altered with fewer transistors compared to previous studies (Table 1).

**Electrical and mechanical stability characteristics**
To ensure reliable and versatile operation, we examined the retention performance of our GMT device (Fig. 5a, b). For the retention

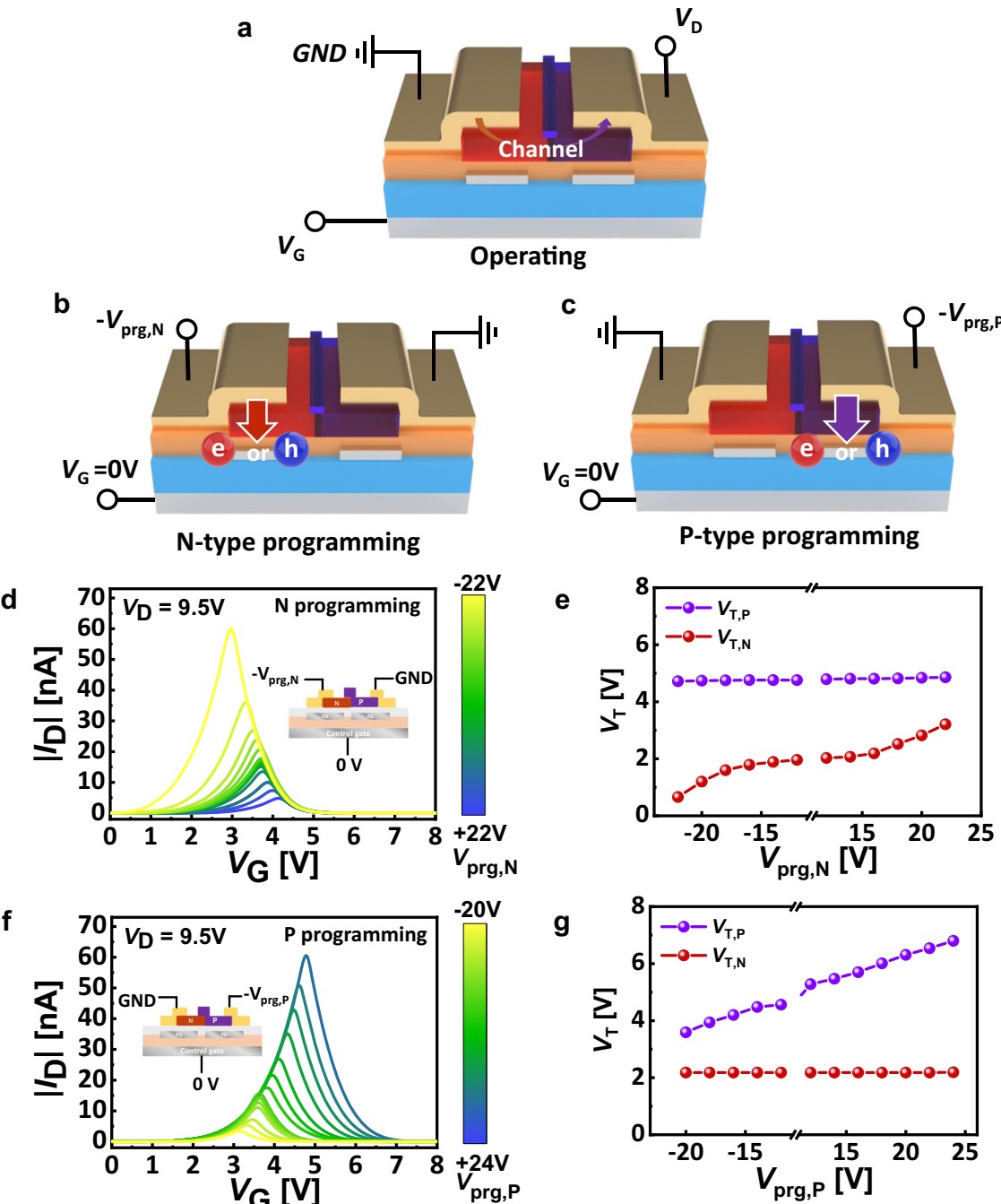

**Fig. 3 | GMT Device programming method. a–c** Schematic illustration of Gaussian-like memory transistor (GMT) device operation (**a**), n-type programming (**b**), and p-type programing (**c**) with the drain voltage ($V_D$), gate voltage ($V_G$), p-type programming voltage ($V_{prg,P}$), and n-type programming voltage ($V_{prg,N}$). **d, e** The change in transfer characteristics (absolute value of drain current ($|I_D|$) versus $V_G$) (**d**) and threshold voltage ($V_T$) according to n-type semiconductor programming (**e**). **f, g** The change in transfer characteristics (**f**) and $V_T$ according to p-type semiconductor programming (**g**).

performance in $\mu$ regulation, we applied $V_{prg,P} = +21\,V$ and $V_{prg,N} = +20\,V$ with 1 s pulse width to induce a sufficient change in $\mu$ value (1.19 V). To assess the retention capability in $\sigma$ regulation, $V_{prg,P} = +16.5\,V$ and $V_{prg,N} = -17\,V$ with 1 s pulse width was applied to double the maximum current value. The changes in $\mu$ and $\sigma$ values over time were extracted from the transfer curves (Fig. 5c). The $\mu$ value at 1000 s, 3000 s, and 10,000 s after programming showed 5.14 V, 5.11 V, and 5.05 V, respectively, which indicates the change in $\mu$ was less than 0.14 V from the initial $\mu$ value (5.19 V) even after 10,000 s. The change in $\sigma$ was also minimal, remaining within 0.016 V of the initial value of 0.440 V (0.433 V, 0.420 V, and 0.418 V at 1000 s, 3000 s, and 10000 s,

respectively). These retention characteristics are attributed to the trapped charge on the isolated AuNP and the minimized leakage path in the dielectric layers[50–52]. For $\mu$ and $\sigma$ cycle endurance measurement, appropriate signs of $V_{prg,P}$ and $V_{prg,N}$ were repeatedly cycled. Remarkably, the GMT device also exhibited cyclic endurance (Supplementary Fig. 16). These retention and endurance characteristics of our GMT device are essential for reliable inference computations[63–65]. The separate FG structure, introduced to control Gaussian-like I-V characteristics, leads to a remarkable reduction of over 50 times in power consumption and latency than other conventional CMOS techniques, when we calculate GMT characteristics emulated with a

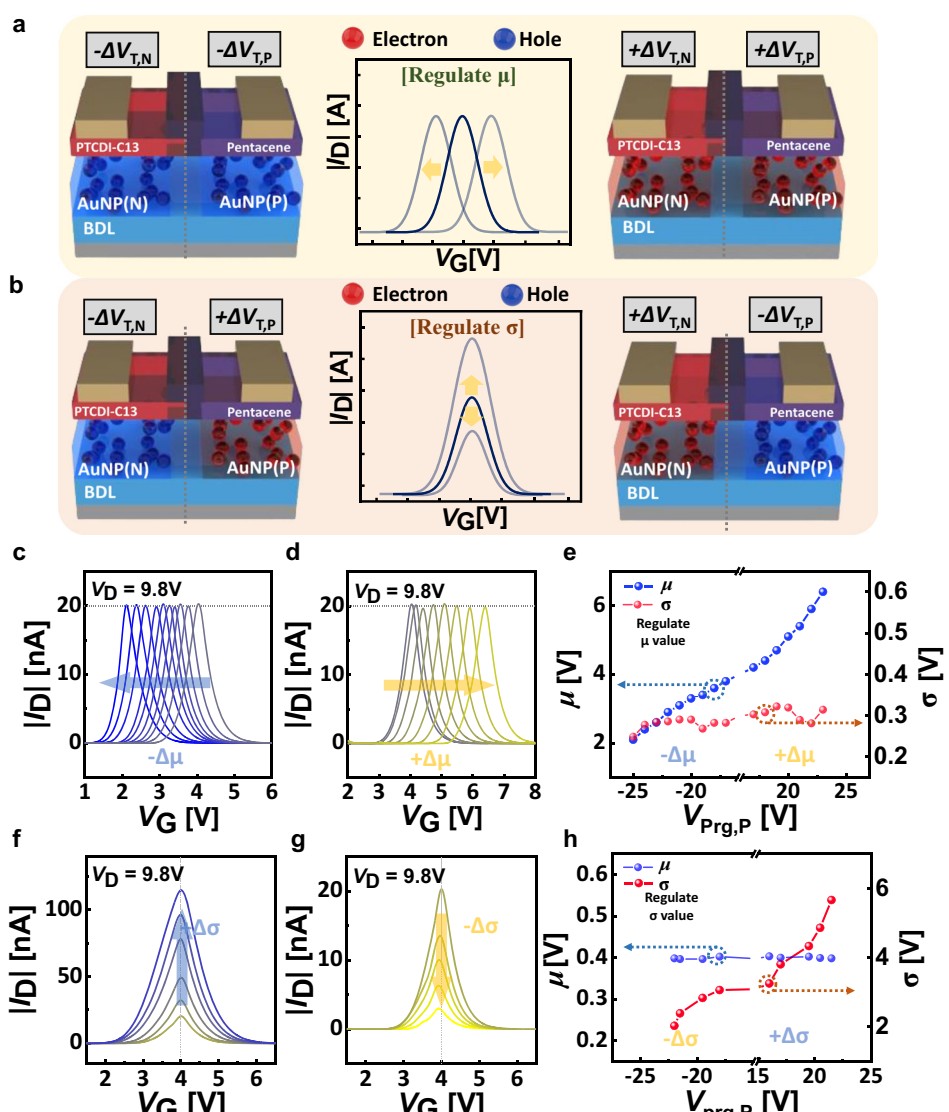

**Fig. 4 | Gaussian factor control of GMT device. a** A schematic illustrations of the same amount of charges with the same polarity injected through the separate floating gate (FG) and corresponding transfer curve shift of Gaussian-like memory transistor (GMT) device. Blue and red symbols represent hole and electron carriers, respectively. **b** A schematic illustrations of the same amount of charges with different polarity injected through the separate floating gate (FG) and corresponding transfer curve shift of GMT device. **c, d** The transfer characteristics (absolute value of drain current ($|I_D|$) versus gate voltage ($V_G$)) measured in Case 1 (movement of p-type threshold voltage ($\Delta V_{T,P}$) = movement of n-type threshold voltage ($\Delta V_{T,N}$)) with the p-type programming voltage ($V_{prg,P}$) < 0 (**c**) and $V_{prg,P}$ > 0 (**d**). **e** The mean ($\mu$) and standard deviation ($\sigma$) values extracted from transfer curves with respect to $V_{prg,P}$ measured in Case 1. **f, g** The transfer characteristics measured in Case 2 ($\Delta V_{T,P} = -\Delta V_{T,N}$) with the condition of $V_{prg,P}$ > 0 (**f**) and $V_{prg,P}$ < 0 (**g**). **h** The mean ($\mu$) and standard deviation ($\sigma$) values extracted from transfer curves with respect to $V_{prg,P}$ measured in Case 2.

45 nm CMOS circuit. Furthermore, the device configuration with just three terminals single transistor contributes to its inherent simplicity, rendering it highly advantageous for future high-density integration.

The proposed GMT devices also offer a promising solution for probabilistic inference in wearable devices and robotics[66], where mechanical stability is crucial to meet various form factors[67–69]. To validate the reliable operation of our GMT device in such applications, we fabricated it on a flexible polyethylene naphthalate (PEN) substrate (Fig. 5d). Figure 5e demonstrates that the Gaussian distribution-like current output remained intact even under the applied tensile strain as high as 2.0%. Furthermore, the GMT device exhibited remarkable electrical stability even after 1000 bending cycles at a fixed applied tensile strain of 1.2% (Fig. 5f). Although marginal current level decrease was observed and the voltage with maximum current point ($V_{max}$) also shifted slightly due to the small dimensional variation by the applied strain, the $I-V$ characteristics fully recovered after releasing from the

tensile strain, supporting that these changes are not caused by device degradation[56,70–73] (Fig. 5g, h). These results confirm the GMT device shows full compatibility with flexible substrates.

**Simulation for insect size-drones mapping model**
Next, we evaluate the performance of GMT-based architectures for probabilistic reasoning for key autonomous navigation objectives, namely, drone localization and obstacle-free path planning[74]. Figure 6a, b show crossbars of GMTs that can physically evaluate likelihood functions for probabilistic localization and path planning, respectively, which use a co-designed map function discussed below, to significantly accelerate their processing.

Compared to CMOS-based digital accelerator, that sequentially computes a likelihood function over hundreds of clock cycles, the architecture of GMT-based crossbar is significantly simplified in the circuit and has increased energy efficiency (Fig. 6a–c). The currently

**Table 1 | Table of the performance of the reported CMOS-based Gaussian I-V characteristics system and GMT device**

| Gaussian Circuit | Gilbert [29] | Delbruck [30] | Adjustable [26] (2021) | This work |
|---|---|---|---|---|
| Circuit structure | | | | |
| Num of TR | 4 | 7 | 14 | 1 |
| Input variable | 3 | 3 | 5 | 1 |
| Regulate factor | $\mu$ | $\mu$ | $\mu,\sigma,A$ | $\mu,\sigma,A$ |

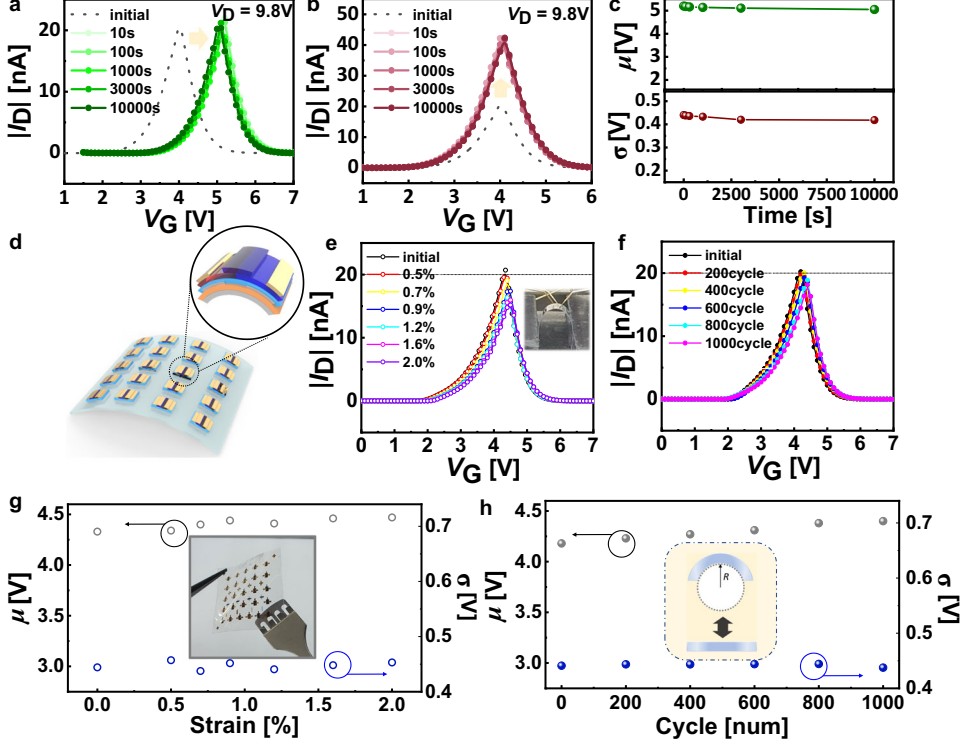

**Fig. 5 | Stability of the GMT device. a**, **b** The change in transfer characteristics (absolute value of drain current ($|I_D|$) versus gate voltage ($V_G$)) of the Gaussian-like memory transistor (GMT) device according to time after optimum programming state of mean ($\mu$) (**a**) and standard deviation ($\sigma$) (**b**) regulation. **c** The change in $\mu$ (green) $\sigma$ value (brown) versus time after the optimum programming state. **d** A schematic illustration of GMT devices fabricated on flexible substrate. **e**, **f** The change in transfer curves according to the applied tensile strain (**e**) and various bending cycles with fixed tensile states strain of 1.2% (**f**). **g**, **h** The variation of $\mu$ and $\sigma$ values according to the applied tensile strain (**g**) and various bending cycles with fixed tensile strain of 1.2% (**h**).

demonstrated GMT devices are sizable and therefore operate at elevated voltages, approximately 10 V. To evaluate the advantages of the scaled GMT-based architectures for probabilistic inference against digital CMOS, we emulated GMT characteristics with a 45 nm CMOS circuit (Supplementary Fig. 18). It is worthwhile to note that increasing the mixture functions in GMT network in Fig. 6a only increases the number of GMT devices but still requires three logarithmic analog-to-digital converters (log-ADCs) and three digital-to-analog converters (DACs). The energy dissipation of the proposed structure for computing the likelihood from a 100-mixture component GMM, in comparison to that of digital data-path is presented in Table 2 and Supplementary Tables 1 and 2. For computing the likelihood, the digital data-path requires thousands of addition/subtraction and comparison, and hundreds of multiplications and read operations. Such extensive workload results into ~941 pJ energy dissipation.

Meanwhile, only a handful of operations are needed for the proposed GMT-based architecture since the Gaussian-like computations can be directly mapped to the characteristics of the device, only requiring ~18.33 pJ energy (Supplementary Table 2). Thus, the proposed scheme requires about 50 times lower energy than the digital data-path. Furthermore, when compared to the state-of-the-art 14 nm and 7 nm digital CMOS, it also exhibits slightly improved performance (Supplementary Table 3).

By leveraging Bayes' rule, the particle filtering method seamlessly integrates single-shot maximum a posteriori (MAP) estimates of robot positions with sequential localization stages (Supplementary Note 7). In particular, we co-designed mapping function for the flying domain from the traditionally used GMM-based map models to better suit the characteristics of GMTs and adapted expectation-maximization procedures to learn the co-designed map models. To better match the

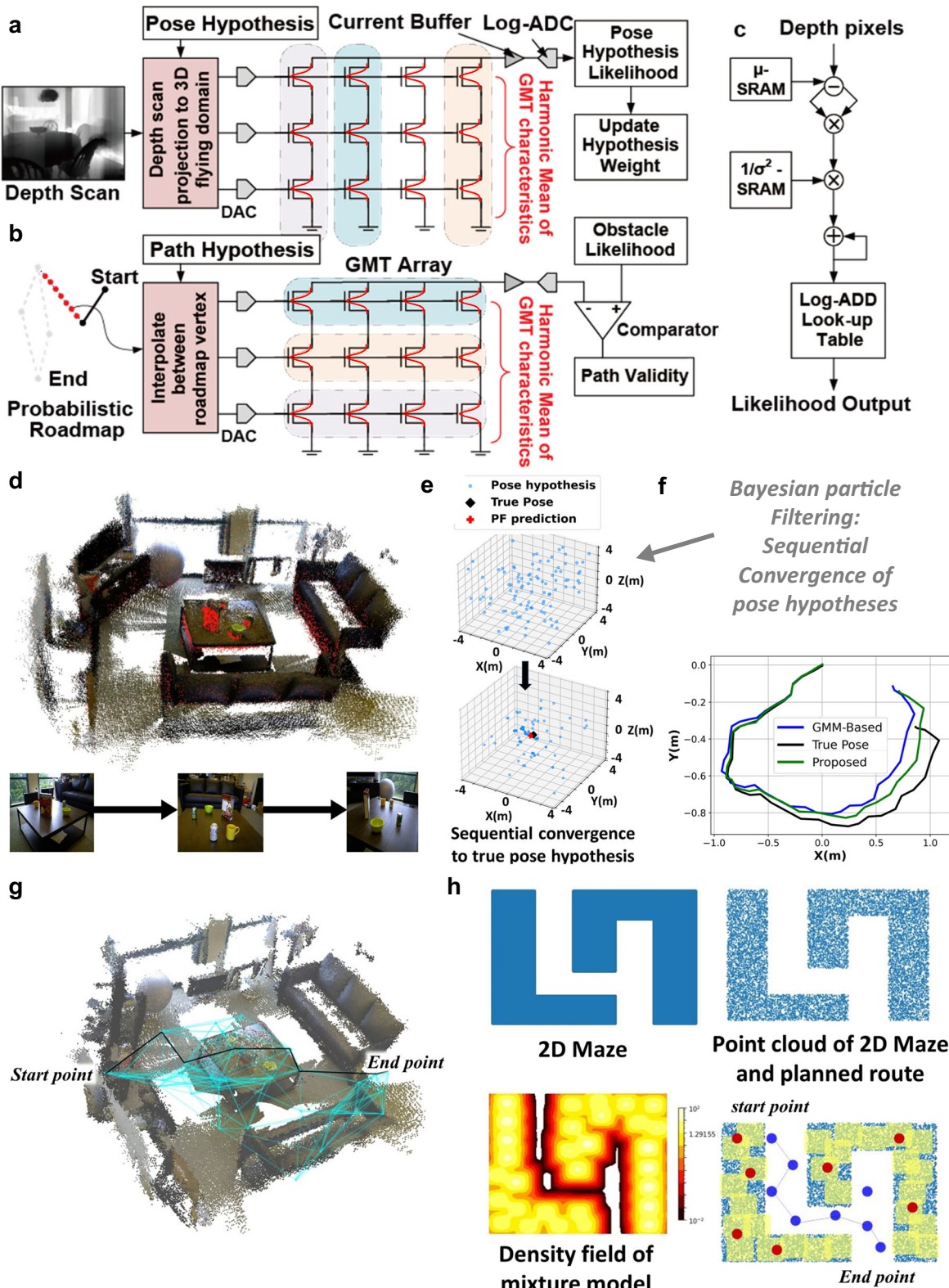

**Fig. 6 | Robotic drone applications based on simulated GMT array. a** Mapping of robotic pose localization to Gaussian-like memory transistor (GMT) array.
**b** Mapping of robotic path planning to GMT array. **c** A comparable digital processing pipeline for evaluating the output of a Gaussian mixture model (GMM)-based statistical mixture function. **d** 3-dimensional (3D) Environment map. **e** Particle filtering in action based on GMT network. **f** Drone position tracking and comparison to ground truth and conventional model. **g** Path planning based on probabilistic roadmap on 3D environment map. **h** A 2D representation of a maze with its point cloud and obstacle density contour as mapped onto GMT network. A schematic drawing of probabilistic roadmap-based path planning is also shown.

**Table 2 | Table of improvement in key performance metrics for likelihood computation using 100-mixture, 3-dimensional density function at 45 nm technology node**

| Metric | Digital Datapath | GMT Architecture | Improvement |
|---|---|---|---|
| Energy/Like-lihood Step | 941 pJ | 18.33 pJ | ~51.3× |

measured characteristics of GMT, an accurate fitting was attempted using the following equation (Supplementary Fig. 6):

$$I_D = I_0 \exp\left(-abs\left(\frac{V_G - V_\mu}{\sigma^\pm(V_\mu)}\right)^\eta\right) \qquad (2)$$

where $I_O$ is the peak current magnitude which is invariant to $V_\mu$ and only depends on the transistor size and $V_D$. $\sigma^+$ is a fitting parameter modeling $I_D$ roll-off when $V_G > V_\mu$. $\sigma^-$ models $I_{DS}$ roll-off when $V_{GS} < V\mu$. $\eta$ is a fitting parameter modeling the power law index of $log$-$I_D$ and $V_G$ dependence. At varying $V_\mu$, $I_D - V_G$ characteristics resemble a skewed Gaussian-like function where the skewness itself is controlled by the peak gate voltage. We modeled such $V_\mu$-dependent, skewed Gaussian-like $I_D$–$V_G$ of the GMT transistor.

Even though each GMT only emulates one-dimensional (1D) Gaussian-like function, the higher-dimensional maps can be efficiently implemented through an array of GMT devices, without requiring complex analog multiplications, by co-designing the map function to GMT characteristics. A series connection of GMTs is considered (Fig. 6a, b). Since the gate-controlled conductance of each GMT follows a Gaussian-like function, resultantly, their column current follows a harmonic mean of Gaussian-like (HMGL) function. HMGL is a multi-dimensional function, matching the dimension of map model, and is given by

$$\text{HMGL}(V_X, V_X, V_X) = \frac{1}{\frac{1}{G(\mu_X, \sigma_X, V_X)} + \frac{1}{G(\mu_Y, \sigma_Y, V_Y)} + \frac{1}{G(\mu_Y, \sigma_Y, V_Y)}} \qquad (3)$$

Here, for Gaussian-like function along the $x$-axis, $\mu_x$ and $\sigma_x$ are the programmed parameters of GMT that are learned using expectation-maximization procedure[75]. $V_X$ is the applied gate voltage. Likewise, Gaussian-like functions along $y$-axis and $z$-axis are defined. Utilizing a mixture of 3D HMGL functions, the 3D map of flying domain is modeled as

$$M(x,y,z) = \sum_{i=1}^{N} \frac{1}{\frac{1}{G(\mu_{Xi}, \sigma_{Xi}, V_X)} + \frac{1}{G(\mu_{Yi}, \sigma_{Yi}, V_Y)} + \frac{1}{G(\mu_{Yi}, \sigma_{Yi}, V_Y)}} \qquad (4)$$

where arbitrary complex maps can be modeled with a sufficient number of HMGL functions.

To compute the log-likelihood of a predictive hypothesis using the mapping model in Eq. (4), the pixel outputs of depth measurements are projected to 3D based on a position and orientation (i.e., pose) hypothesis for the localizing drone. The transformed outputs are converted to analog domain and applied to the crossbar using digital-to-analog converters (DAC). The output current induced by the crossbar follows the log-likelihood of pose hypothesis as defined by Eq. (4). The log-likelihood output of multiple pixels of a depth map are summed to compute the net likelihood of a hypothesis. The computations are iterated over all hypotheses to decimate the less likely ones.

The hypothesis was projected onto the GMM-modeled domain map and the pose hypotheses converge based on this projected map (Fig. 6d, e). Using this as a foundation, the weighted mean trajectory of the drone position was derived from the considered hypothesis set in

comparison with the ground truth (Fig. 6f). Significantly, our GMT-based co-designed approach fulfils a comparable performance despite various non-idealities such as skewed mapping of a Gaussian-like function and limited programmability of only mean variables of the characteristics.

Likewise, the point cloud map of obstacles in a room was fitted with the co-designed map model. Utilizing the GMT array, a graph of obstacle-free paths is prepared by initially randomly sampling points within the map domain. To verify the validity of the travel path between two adjacent points, the sampled points were interpolated in 3D and applied to the GMT array, while the array's response output current was characterized. If the response current from the array remains below a small threshold for all interpolating points between the sample points, this denotes a very low likelihood of an obstacle present between these two points and the edge connecting them represents a valid path. Figure 6g presents the final graph representation of the potential path through the 3D point cloud map of obstacles. To enhance comprehension, Fig. 6h demonstrates path planning in a 2D environment. Based on the generated graph, the optimal obstacle-free path with the least distance between two locations can be determined using Dijkstra's algorithm[76,77].

The simulation results demonstrate that implemented GMT arrays to obtain probabilistic likelihood functions allow for the analysis of dense graphs using much fewer node devices and reduced computational workload, thus enabling result optimization with rapid operation. Consequently, this approach holds the potential for mitigating overconfidence issues, by operating on probabilistic reasoning models with significantly enhanced both area and energy efficiency.

## Discussion

A device architecture, called the GMT, is proposed, incorporating a heterojunction of p-type and n-type organic semiconductors along with a non-volatile flash memory structure. This conjunct structure allows for the systematic programming of Gaussian distribution-like $I − V$ characteristics from a single device, making it possible to implement probabilistic inference in ultralow-power hardware and highly simplified circuit design. The separate FG memory structure enables independent programmability of the p-type and n-type channel conductance of the GMT device. By programming and erasing each FG, meticulous control of the amplitude ($A$), mean ($\mu$), and standard deviation ($\sigma$) of the transfer curve output of the GMT device is achieved. This controllability, enabled by the separate FGs, not only facilitates the hardware implementation of probabilistic inference but also enhances model circuit feasibility while reducing power consumption and latency. The GMT devices also exhibit remarkable retention performance and cyclic endurance. Moreover, the GMT devices maintain their performance even with 2.0% of the applied tensile strain. Using the GMT device array and by co-designing the probabilistic inference model to device characteristics, we demonstrate applications towards 3D probabilistic localization and path finding for drones. The GMT device, featured by a simple unit transistor design with a heterojunction channel, is capable of representing output values in the form of a probability distribution function. Moreover, the separate floating-gate structure allows for the programmable modification of the probability distribution function without the need for additional components. Furthermore, the reconfigurable device design in this study has a great potential for the application to various probabilistic inference computing fields with increased computational efficiency and integration density thereof. The successful implementation of inference operations is indeed demonstrated by using actual data obtained from GMT devices. Therefore, the proposed heterojunction-based, Gaussian-like transistor embedding a separate floating gate memory structure can serve as a powerful platform for Bayesian operations. For instance, they can improve the robustness and scalability of speech recognition models

that use GMMs and HMMs by efficiently processing large-scale GMMs. In natural language processing, GMTs can accelerate tasks such as word embedding and semantic analysis, thanks to Gaussian function-based techniques. Additionally, GMTs can expedite variational inference, Gaussian processes, and other algorithms by leveraging Gaussian functions to model uncertainties, making them valuable for Bayesian analysis, regression, classification, and optimization in fields like robotics, time-series forecasting, and spatial data analysis.

## Methods

### Device fabrication and characterization
For the fabrication of the GMT device, a 25 mm × 25 mm glass substrate (Samsung Corning Co.) underwent a cleaning process involving ultrasonication in deionized (DI) water, acetone, and isopropyl alcohol for 20 minutes, followed by drying with dry $N_2$ gas. The metal electrode and organic semiconductor were deposited via thermal evaporation in a vacuum of $2 \times 10^{-6}$ Torr, with the thickness monitored in real-time using a quartz crystal microbalance (QCM). The gate electrode, consisting of thermally evaporated Al with a deposition rate of $0.1 \, nm \, s^{-1}$, reached a thickness of 50 nm. The dielectric layers were deposited using an initiated chemical vapor deposition (iCVD) process. A high-k poly(2-cyanoethyl acrylate-co-diethylene glycol divinyl ether) [p(CEA-co-DEGDVE)] with optimized chemical composition (referred to as pC1D1) served as the blocking dielectric layer (BDL)[56], while a low-k poly(1,3,5-trivinyl-1,3,5-trimethyl cyclotrisiloxane) (pV3D3) was used as the tunneling dielectric layer (TDL)[54]. The thicknesses of pC1D1 (BDL) and pV3D3 (TDL) were 100 nm and 14 nm, respectively. Additionally, gold (Au) was thermally evaporated with a deposition rate of $0.01 \, nm \, s^{-1}$ to a thickness of 3 nm for use as the floating gate (FG) between the two dielectric layers[50]. Organic semiconductors, including pentacene and PTCDI-C13, were also thermally deposited at a rate of $0.03 \, nm \, s^{-1}$. PTCDI-C13 underwent recrystallization through thermal annealing at 160 °C for 1 h. For the source and drain electrodes, 70 nm-thick Au was thermally evaporated at a rate of $0.1 \, nm \, s^{-1}$. The channel dimensions were 500 μm for width and 600 μm for length. Flexible GMT devices were fabricated on a 100 μm-thick PEN substrate, and the applied tensile strain was calculated using the following equation[78]:

$$S = \frac{d_{sub}}{2R + d_{sub}} \tag{5}$$

where $S$ is the tensile strain, and $R$ and $d_{sub}$ are the bending radius and substrate thickness, respectively. To obtain cross-section image of GMT device, the device was sliced by a focused ion beam (Helios Nanolab 450) and cross-sectional HRTEM images were obtained by using Cs-corrected TEM (Titan cubed G2, FEI) with EDS mapping analysis. The scanning probe microscope (XE-100, Park Systems) was used to obtain AFM images to analyze the surface morphologies with the scan size of 5 μm × 5 μm. To analyze the insulating properties of polymer dielectric layers, metal-insulator-metal (MIM) devices were fabricated where the polymer dielectric layers were deposited between thermally evaporated Al electrodes. The electrical characteristics were measured by B1500A semiconductor analyzer (Agilent Technologies). All the device fabrication and characterization were performed in the $N_2$-filled glovebox.

### Deposition of polymeric thin film
pV3D3 and pC1D1 dielectric layers were deposited by using a custom-built iCVD reactor. V3D3 (95%, Gelest, USA), CEA ( > 95%, Tokyo Chemical Industry (TCI), Japan), and DEGDVE (99%, Sigma-Aldrich, USA) acted as monomers, while tert-butyl peroxide (TBPO, 98%, Aldrich, USA) served as the initiator. All chemicals were utilized as received, without any additional purification steps. For the deposition of pV3D3, 40 °C heated V3D3 and 30 °C heated TBPO were introduced into the

chamber at flow rates of 2.5 standard cubic centimeter per minute (sccm), and 1 sccm, respectively. The chamber pressure was maintained at 100 mTorr, and the substrate temperature was set to 50 °C[54]. In the case of pC1D1, CEA, DEGDVE, and TBPO were heated to 50°C, 50°C and, 30°C, respectively, and they were injected into the chamber at flow rates of 0.28 sccm, 0.28 sccm, and 0.48 sccm, respectively. The chamber pressure and substrate temperature were maintained at 60 mTorr and 30 °C, respectively[56]. Additionally, the filament was heated to 130 °C to initiate the decomposition of the initiator into radicals. The thickness of the resulting polymer films was determined using a spectroscopic ellipsometer (M2000, Woollam).

## Data availability
The data within the article and its supplementary Information are available from the corresponding authors upon request.

## Code availability
The Code within the article and its supplementary Information are available from the corresponding authors upon request.

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

## Acknowledgements

This work was supported by Samsung Advanced Institute of Technology (SAIT), SEC (S. G. I.). This work was supported by the National Research Foundation of Korea(NRF) grant funded by the Korea government(MSIT), No. 2021R1A2B5B03001416 (S. G. I.), NRF-2022R1C1C1004590 (H. Y.) and RS-2023-00210194 (H. Y.). This work was supported by the Wearable Platform Materials Technology Center (WMC) funded by the National Research Foundation of Korea(NRF) Grant by the Korean Government (MSIT), NRF-2022R1A5A6000846 (S. G. I.). This work was also supported by a grant from National Science Foundation (NSF), grant # 2106824.

## Author contributions

C.L., J.C., H.Y., and S.G.I. conceived the idea and designed the experiments. Chang.L. and J.C. designed, fabricated, and measured all the devices and circuits. J.P. assisted with device characterization, and C.L. helped with the circuit design. S.M.L. assisted device fabrication. J.C., C.L., H.Y., and S.G.I. wrote the manuscript. L.R., D.K., P.S., and A.R.T. conceived the application-level ideas. L.R., D.K., and P.S. simulated the design. A.R.T. prepared the description of the results. All authors reviewed the manuscript and discussed the results. C.L., J.C., and L.R. contributed equally to this work. A.R.T., H.Y. and S.G.I. contributed equally as corresponding authors.

## Competing interests

The authors declare no competing interests.
