## [Peer Review File · Nature Communications]

Highly Parallel and Ultra-Low-Power Probabilistic Reasoning with Programmable Gaussian-like Memory TransistorsREVIEWER COMMENTS

Reviewer #1 (Remarks to the Author):

Dear Authors,

I believe it is in an author's best interest to receive a review with the highest possible amount of fair criticism, as this associates their name with high-quality work. Despite time constraints, I dedicated the maximum amount of time I could to this review and aimed to be as critical as possible.

As a reviewer, my general opinion is that the topic is interesting and warrants publication after major/minor revisions.

Below, you'll find my complete review, which includes both major and minor issues:

1) Please provide more related works. Consider checking Google Scholar for "Gaussian Mixture Model hardware" from 2019 and earlier. This addition will enhance the overall quality of your work.

2) Kindly offer a more specific and accurate abstract, as the current version is somewhat general.

3) Could you clarify the main advantage of your work in comparison to the existing literature? While you implement a GMT with the advantage of tuning Gaussian curve characteristics via FG, it's apparent to the reader that the VD should be up to 11V. This implies a high-power supply rail. Please provide further clarification.

4) How does your work address high voltage supply rails, particularly in wearable applications? How does your circuit manage this scenario?

5) I recommend splitting the "Introduction" section into two distinct sections: a) Introduction, and b) Literature Review. This adjustment will facilitate reader comprehension.

6) A thorough grammar and English check is necessary. For instance, in Figure 2, "stricture" is a different word than "structure." Please rectify this.

7) As readers can observe, in the literature (reference 28), the characteristics of the Gaussian curve are independent. It is not clear from Figures 3 and 4 whether this also holds true for your work. Please provide clarification.

8) Have you explored whether this method can be combined with a different ML model? I recommend consulting related literature. For example, please refer to the following work to understand the use of GMT in different classifiers: "General Methodology for the Design of Bell-Shaped Analog-Hardware Classifiers."

9) Have you evaluated your GMT over corners or in the presence of PVT variations?

10) Please include a comparison table with related works and corresponding metrics. This addition will enhance the overall quality of your work.

11) I suggest improving the text size in the main figures, as it appears too small in certain cases.

12) Please refine the way you articulate your work. Utilize third-person language rather than first-person. For instance, instead of saying "We believe that we can offer a fundamental platform," explain why you can offer this fundamental platform and support it with your results.

Thank you for your attention to these suggestions. I look forward to seeing the revisions.

Reviewer #2 (Remarks to the Author):

Manuscript "Highly Parallel, Ultra-Low-Power Probabilistic Reasoning with Programmable Gaussian Memory Transistors" by Lee et al describes an experimental realization of a simple transistor-like device that produces a Gaussian-like response with controllable mean and variance. Although there are other realizations of a Gaussian-like response in analog devices, but they require a fairly complicated circuit; the demonstrated device thus can greatly decrease the required power requirements. Utility of networks of such devices for 3-dimensional data analysis/learning tasks has also been demonstrated. This is an impressive achievement, and will be of interest to the broad readership of Nat. Comm. However, I would like the authors to clarify/address several issues before the manuscript is published.

- Experimental realization of GMT in this work used organic semiconductors. Then potential performance advantages were modeled assuming 45nm CMOS realization of GMT. Better motivation and more details are needed to place such estimates in proper context. For example, why 45nm were assumed? Is it because the floating gate transistors cannot be realized using current state-of-the-art feature sizes (<10nm)? In any case, the comparison of GMT array-based vs digital realization of Gaussian Mixture Model should use state-of-the-art for the latter (<=5nm). Also, speed/latency comparison is unrealistic since all modern (at least, high-power) digital processors are data-parallel hence latency modeling needs to take that into account.

- The manuscript uses mixture of Gaussian vs Gaussian-like to characterize the device. The output is clearly not Gaussian. Even though in the analog community such response is modeled by Gaussians, that's a crude approximation; it might be OK for extremely crude tasks where only a few bits of information are available, but for other tasks of interest to the readership of Nat. Comm. (e.g., physical simulation) the device response will need to be modeled more precisely. Intuitively it seems that the Gaussian-like response is obtain via competition of 2 channels so it should be modeled well as a product of 2 Fermi-Dirac distributions with opposite signs of variables and 2 different "variances". Indeed, playing around with such distribution seems to produce shapes that look more like the asymmetric responses measured here for GMT than the produced Gaussian fits. The tails of the distributions are clearly *exponential* (i.e., $\exp(-k x)$) not Gaussian $\exp(-k x^2)$. So clearly describing the device response as "Gaussian" is a stretch, and the device should be termed "Gaussian-like" Memory Transistor. Consequently, I suggest that the authors revisit the awkward design of the function in Eq. (2); at the very least there needs to be a comparison of use of Eq. (2) vs. regular Gaussian. P.S. Of course, for the crude learning tasks the differences do not matter, but this points out that using Gaussians might be suboptimal for the implementation on a digital device and functions that are more economically

implementable can be used. The authors might want to comment on this.

- Equation (3) is extremely confusing to a non-specialist. Single product of sums of 1-d Gaussians can only model signal in a small domain such that the signal is very smooth. In a larger 3-d domain the signal must be approximated as a linear combination of 3-d Gaussians. Clearly, this is what is done in practice here, as Figure 6h suggests; this needs to be clarified.

Response to the Reviewers' Comments

The authors thank the Reviewers for their considerate review of our manuscript and the valuable comments. We have revised the manuscript and supporting information based on the Reviewers' comments. The Reviewers' comments appear in **black**, and the authors' responses in blue. In the revised manuscript, **the changes** with respect to the previous version are highlighted in yellow.

Reviewer (#1)'s COMMENTS:

I believe it is in an author's best interest to receive a review with the highest possible amount of fair criticism, as this associates their name with high-quality work. Despite time constraints, I dedicated the maximum amount of time I could to this review and aimed to be as critical as possible.

Response:

We appreciate the Reviewer #1 for your considerate in-depth review of our manuscript and providing the valuable comments. We have carefully revised the manuscript by preparing the point-by-point response to each comment as follows.

Comment #1) Please provide more related works. Consider checking Google Scholar for "Gaussian Mixture Model hardware" from 2019 and earlier. This addition will enhance the overall quality of your work.

Response:

We fully agree with Reviewer #1's suggestion to include information on research related to utilizing Gaussian mixture model (GMM) for various prediction tasks. The GMM has been applied in a versatile manner to undertake various predictive tasks such as speech processing^[R1-2], classification^[R3], density estimation^[R4], and so on. Given the importance of these models in real-time and resource-efficient deployment for such applications, several prior works have developed analog and digital hardware for the models. Minghua *et al.* explored digital very large scale integrated circuit (VLSI) implementation of GMM by linear piecewise approximation of Gaussian functions.^[R5] Some previous studies also explored application-specific integrated circuit (ASIC) and field-programmable gate array (FPGA) implementations of the model for

real-time tasks such as segmentation of high-definition video.^[R6-8] The digital implementations, however, required considerable area and power consumption to limit scalability for low-power, high-performance tasks. A vote count-based in-memory GMM implementation on a dynamic random-access memory (DRAM) was also presented.^[R9] However, the developed platform was limited to classification tasks and was not extended to regression frameworks. Ultralow-power analog circuits for GMM-based seizure prediction had also been presented in the previous work,^[R10] nonetheless, each Gaussian cell required twelve transistors. On the other hand, the platform developed in this study successfully implements GMM within a single device and along with variance programmability, as well as the extendibility to regression frameworks. We added the following new references to our work.

We revised the manuscript as follows;

Page 4, 1st paragraph,

Notably, the computation of a GMM's output entails a multitude of multiplications, additions, and look-up operations. Such workload in computation grows with the signal dimension and the complexity of the model, determined by the number of mixture functions in the GMM²⁴. Given the importance and versatility of GMM, previous studies have investigated GMM computation hardware intensively utilizing dynamic random-access memory (DRAM),²⁵ field-programmable gate array (FPGA),^{26,27,28} or analog circuits. However, usually a large number of transistors was required, which inevitably had resulted in a high level of complexity.²⁹ Consequently, without innovative technologies that can dramatically reduce resource demands when dealing with high-dimensional complex distribution functions like GMMs, the advantages of probabilistic inference procedures remain out of reach for numerous applications.

Page 4, 2nd paragraph,

For instance, analog circuits based on silicon-based complementary metal–oxide–semiconductor (CMOS) technology have been used to emulate Gaussian functions. However, in conventional Si metal-oxide-semiconductor field-effect transistor (MOSFET)-based circuits, at least four transistors are required to represent a Gaussian distribution.^{29, 30} Michail *et al.* presented a GMM-based classifier by connecting NMOS and PMOS bump circuits with a winner-takes-all (WTA) circuit.³⁶ Another study, Vassilis *et al.* proposed low-power bell-shaped

analog classifiers (CLFs) by implementing GMM (machine learning) circuits with fewer number of transistors and three input values, and tested on real-world biomedical classification problems.³⁷ Nevertheless, to tune the characteristics of the distribution function, the number of transistors increases often to more than 10, rendering the circuit quite complex and inevitably increasing power consumption.

[R1] *IEEE Transactions on Speech and Audio Processing* 1995, 3, 1, 72-83.

[R2] *IEEE International Conference on Acoustics, Speech and Signal Processing*, 2010, 4334-4337

[R3] *Journal of Statistical Planning and Inference* 2010, 140, 5, 1175-1181.

[R4] *Computational Statistics*, 2013, 28, 127-138.

[R5] *IEEE Transactions on Very Large-Scale Integration (VLSI) Systems* 2006, 14, 9, 962-974.

[R6] *IEEE International Conference on Electronics, Circuits and Systems*, 2006, 1276-1279.

[R7] *IEEE Transactions on Very Large-Scale Integration (VLSI) systems* 2013, 22, 3, 537-547.

[R8] *IEEE Transactions on Computers*, 2017, 66, 11, 1837-1850.

[R9] *IEEE Signal and Information Processing Association Annual Summit and Conference*, 2014.

[R10] *Bioengineering* 9, no. 4 (2022): 160.

Comment #2) Kindly offer a more specific and accurate abstract, as the current version is somewhat general.

Response:

Thank you very much the Reviewer #1 for pointing out the ambiguity in abstract. In the revision, we meticulously reviewed the abstract to revise ambiguous expressions. Furthermore, for specific and accurate abstract, we added quantitative information of the Gaussian-like memory transistor (GMT) device.

The revised abstracts are as follows:

Probabilistic inference in data-driven models has garnered critical interest for its promising capability to predict outputs and associated confidence levels, alleviating risks arising from overconfidence in learning and sensing. However, implementing complex statistical computations in hardware with minimal devices still remains challenging. Here, utilizing a heterojunction of p- and n-type semiconductors coupled with separate floating gate configuration, a Gaussian-like memory transistor (GMT) is proposed, where a programmable Gaussian-like current-voltage response is produced within a single device. A separate floating-gate structure is incorporated into an anti-ambipolar heterojunction transistor, which allows for the exquisite control of the Gaussian-like current output, including mean (μ) and standard deviation (σ), which enables physical evaluation of complex distribution functions. The GMT could effectively modulate the Gaussian-like distributions to a significant extent through simple programming, with the μ position changed by 200% of the width (set as 6σ according to 3σ rules) and σ changed by 100% of the initial value. Our GMT device also features excellent retention performance, (μ and σ values change by only 140 mV and 16 mV, respectively, after 10^4 s) as well as superior mechanical flexibility (fully operational even under 2% strain). The programmable 3-terminal single transistor GMT structure simplifies circuit design for probabilistic inference and achieves higher parallelism. Successful implementation of probabilistic inference for localization and obstacle avoidance tasks is demonstrated using Gaussian-like curves from GMT devices. With its ultralow-power consumption, simplified design, and the capability to produce programmable Gaussian-like outputs, the 3-terminal GMT device developed in this study has the promising potential as a hardware platform for probabilistic inference computing.

Comment #3) Could you clarify the main advantage of your work in comparison to the existing literature? While you implement a GMT with the advantage of tuning Gaussian curve characteristics via FG, it's apparent to the reader that the V_D should be up to 11 V. This implies a high-power supply rail. Please provide further clarification.

Response:

We appreciate Reviewer #1's valuable comment. We fully agree with Reviewer #1 in that the fabricated GMT device requires a relatively high driving voltage. Meanwhile, the real value of our study lies in the novel structure of devices and the operating scheme that have the potential to improve substantially the computational efficiency and integration density of probabilistic inference computing devices. Through this paper, we would like to emphasize the need for concurrent research and advancement in such novel device architecture to reducing the number of devices.

The drain voltage (V_D) value of 11 V was required only during the analysis to verify the change in the Gaussian-like characteristics with respect to V_D . In fact, all operations were conducted at a voltage below 9.5 V, and the gate voltage no larger than 8 V. Nevertheless, the driving voltage is somewhat higher compared to current silicon industries. The main reason for the relatively high driving voltage is because the device is based on organic materials, which have suffered from 1) the absence of high-performance insulating materials and 2) the relatively lower mobility of organic semiconductors.

In addition, in order to demonstrate a programmable Gaussian-like current (I)–voltage (V) characteristics, the flash memory structure was implemented, where the insulating layer must maintain its characteristics without insulating breakdown even under the high field applied (usually at least greater than 5 MV/cm) during the programming process. At the same time, because the operating voltage of the device is directly related to the capacitance (C_i) of the insulating layer, an insulating layer with high dielectric constant is required, and it should maintain superb insulation performance even at a low thickness, to increase C_i value.

Such a low-voltage, stable memory operation has been extremely challenging especially in organic insulating materials. As depicted in Fig **R1**, most of the organic insulating layers usually require the thickness at least higher than 100 nm to secure the sufficient level of dielectric robustness. On the other hand, the insulating layers based on initiated chemical vapor deposition

(iCVD) have exhibited one of the best insulation performances among the reported organic insulating materials, and they are practically the only materials that have the comparable electrical performance compared to inorganic ones,^[R11-R20] which allowed for relatively low operating voltages in our organic semiconductor-based GMT devices.

Fig. R1. The dielectric constant (k) vs thickness plot of various organic materials.

The dielectric constant (k) vs thickness plot of iCVD dielectric materials compared to those of the single polymer dielectrics developed in previous reports. (Dot line indicate the effective oxide thickness (EOT))

In this study, exploiting iCVD process, we utilized poly(2-cyanoethyl acrylate-co-diethylene glycol divinyl ether) [p(CEA-co-DEGDVE)] blocking layer, demonstrating excellent insulation performance with a dielectric constant (k)-value of higher than 6. Additionally, a thin poly(1,3,5-trivinyl-1,3,5-trimethyl cyclotrisiloxane) (pV3D3) tunneling layer film was introduced on the upper part of the floating gate. At the same time, to ensure reliable memory

operation, we aimed to achieve a reliable level of insulation by utilizing an insulation layer with a sufficient thickness of 100 nm for the blocking layer of p(CEA-co-DEGDVE) and 14 nm for the tunneling layer of pV3D3.

To address the Reviewer #1's comment, we optimized insulating layer thickness to withstand the required field without dielectric breakdown. Consequently, we developed a device utilizing 70 nm p(CEA-co-DEGDVE) blocking layer and 8 nm pV3D3 tunneling layer. Using the GMT device with the reduced insulating layer thicknesses, with $V_D = 4.5$ V, we observed output current values comparable to those obtained with $V_D = 10.5$ V in our original GMT device. The gate voltage was further scaled down to less than 3 V. Furthermore, when programming the fabricated devices, we successfully achieved the systematic movement of a Gaussian-like-shaped graph without device failure. (Fig R2). The additional scaling down of operating voltage strongly suggested the potential for the demonstration of ultralow-voltage GMT devices, while maintaining their applicability to flexible electronics.

Fig. R2. The electrical characteristics of newly fabricated GMT device.

a,b, The C–V characteristics of BDL (Al/p(CEA-co-DEGDVE)/Al MIM device) (a) and TDL (Al/pV3D3/Al MIM) (b). **c,** The transfer characteristics newly fabricated GMT device. **d,** The transfer characteristics measured in parallel movement programming (mean (μ) value change).

Besides, concerning the semiconductor materials, we employed organic small molecules, *N,N'*-ditridecyl-3,4,9,10-perylenetetracarboxylic diimide (PTCDI-C13) and pentacene for the n-type and p-type semiconductors, respectively. To maximize the performance of the currently employed organic semiconductor, two approaches can be suggested. First, organic thin-film transistors (OTFTs) typically suffer from electron state decoupling between the metal and molecules due to van der Waals (vdW) metal-molecule contact, which often causes high contact resistance.^[R21] Therefore, a reduction in contact resistance can be achieved through the introduction of contact doping^[R22] or the incorporation of a charge injection layer between the organic semiconductor and the contact metal.^[R23] This approach can enhance the performance of organic transistors. Second, the charge transport characteristics of OTFTs are mainly constrained by the low crystallinity of organic semiconductors and numerous crystalline-grain boundaries, resulting in charge trapping and low intrinsic mobility within the channel.^[R24] Various methods, such as molecular engineering^[R25-26], surface modification^[R27], and deposition technique improvements^[R28], can be employed to enhance the intrinsic mobility within the channel. Through these strategies, molecular orientation in organic semiconductors can be controlled, forming well-ordered molecular packing structures to further improve the intrinsic mobility within the channel and reduce the operating voltage of the device.

Importantly, it is worth noting that the proposed device design is not strictly confined to specific material systems, as long as the channel layer can produce anti-ambipolar characteristics by constructing a sharp, reliable p-n heterostructure. Therefore, by utilizing materials that exhibit both p-n junction operation and the improved charge transport characteristics, it is expected to achieve sufficient current at low operating voltages. For example, in previous research, the utilization of highly mobile 2D materials such as MoTe₂ and MoS₂ allowed achieving a comparable current level with a relatively low V_D voltage. Meanwhile, the gate voltage remained high due to the presence of a thick insulating layer.^[R29] Furthermore, utilizing single-wall carbon nanotube (SWCNT) and amorphous indium gallium zinc oxide (*a*-IGZO) with a thin hafnia (HfO₂) insulating layer of 15 nm resulted in not only a relatively low V_D at 2 V but also an exceptionally low voltage output for gate driving within 3 V.^[R30] However, we believe demonstrating devices using other semiconductor materials and their heterojunction is beyond

the scope of this study and it would be more appropriate to present them in a follow-up study with in-depth analysis.

We revised the manuscript as follows;

Page 7, 2nd paragraph,

As shown in Fig 2d, a Gaussian distribution-like current output was indeed implemented in the GMT device, regardless of different drain voltages (V_D). The operating voltage of the GMT devices could be reduced to less than 3 V with further scaling down the thickness of dielectric layers. (Supplementary Fig. 17). With the variation in V_D , only off voltage (V_{off}) was affected, while on voltage (V_{on}) remained constant, causing the variation of V_{off} with the change in V_D (Supplementary Fig. 6)^{50, 51}.

Page 16, 2nd paragraph,

Using the GMT device array and by co-designing the probabilistic inference model to device characteristics, we showed applications towards 3D probabilistic localization and path finding for drones. The GMT device, featured by a simple unit transistor design with a heterojunction channel, is capable of representing output values in the form of a probability distribution function. Moreover, the separate floating-gate structure allows for the programmable modification of the probability distribution function without the need for additional components. Furthermore, the new reconfigurable device design in this study has a great potential for the application to various probabilistic inference computing fields with increased computational efficiency and integration density thereof. The successful implementation of inference operations was indeed demonstrated by using actual data obtained from GMT devices. Therefore, the newly proposed heterojunction-based, Gaussian-like transistor embedding a separate floating gate memory structure could serve as a powerful platform for Bayesian operations. For instance, they can improve the robustness and scalability of speech recognition models that use GMMs and HMMs by efficiently processing large-scale GMMs. In natural language processing, GMTs can accelerate tasks such as word embedding and semantic analysis, thanks to Gaussian function-based techniques. Additionally, GMTs can expedite variational inference, Gaussian processes, and other algorithms by leveraging Gaussian functions to model

uncertainties, making them valuable for Bayesian analysis, regression, classification, and optimization in fields like robotics, time-series forecasting, and spatial data analysis.

We added the electrical characteristics of newly fabricated GMT device (Fig **R2**) to Supplementary Fig **17** in Supplementary Information (p. 21)

- [R11] *Org. Electron.*, 9 (2008) 1069–1075
- [R12] *Nat. Nanotechnol.*, 2007,2, 784–789
- [R13] *Nature*, 2009,457, 679–686.
- [R14] *J. Appl. Phys.* 2004,95, 1594–1596
- [R15] *Chem. Mater.* 2010, 22, 1559–1566
- [R16] *Org. Electron.* 2016,36, 171–176.
- [R17] *Polym. Chem.*, 2015, 6, 3685–3693
- [R18] *Adv. Electron. Mater.* 2016, 2, 1500209
- [R19] *Nat. Mater.*2015,14,628–635.
- [R20] *ACS Appl. Mater. Interfaces.* 2017, 9, 20808–20817
- [R21] Simon M. Sze, *Physics of semiconductor devices*. John wiley &sons, 2021,
- [R22] *Chem. Rev.*, 2016, 116(22), 13714-13751.
- [R23] *Adv. Mater.*, 2022, 34(2), 2104075.
- [R24] *Chem. Soc. Rev.*, 2010, 39(7), 2643-2666.
- [R25] *Nat. Rev. Chem.*, 2020, 4(2), 66-77.
- [R26] *Adv. Funct. Mater.*, 2022, 32(21), 2200843.
- [R27] *Adv. Mater.*, 2010, 22(34), 3857-3875.
- [R28] *Energy Environ. Sci.*, 2014, 7(7), 2145-2159.
- [R29] *Appl. Phys. Lett.* 2020, 117, 123103
- [R30] *Nano Lett.* 2015, 15, 416–421

Comment #4) How does your work address high voltage supply rails, particularly in wearable applications? How does your circuit manage this scenario?

Response:

We are grateful for the constructive comment provided by Reviewer #1. As the Reviewer #1 pointed out, due to a larger dimension of fabricated transistors, our laboratory-scale devices require ~10 V to operate which would require handling high voltage supply rails, which has quite a room for improvement, as addressed in the response to Comment #3 above. As presented in Fig **R2**, we have indeed identified the potential to achieve an operating voltage of less than 5 V through the downscaling of the dielectric thicknesses. The reduced operating voltage ranges are fully acceptable for wearable electronics, as demonstrated in the previous research where wearable devices attached on human skin with the red organic light-emitting diode (OLED) array were successfully implemented with electronic component with a driving voltage of 5 V and a 3.7 V lithium-ion battery.^[R31] In fact, for wearables applications, we would like to point out that even with the current adaptation, the 10 V operation of GMT falls within safety margins recognized for human health.^[R32]

Meanwhile, we fully agree with the reviewer's concern that a higher voltage operation would be necessary for writing on the floating gate of GMT than in typical logic and analog circuits. However, we also expect that with GMT's gate stack scaling, these programming voltages will proportionally shrink to typical input/output (I/O) voltage (3.3 – 5 V) range and can be handled by typical I/O circuits such as level shifters and programming circuits for flash memories. Even more, the proposed non-von Neumann processing significantly minimizes the necessary write operations to begin with. A write operation is needed only to store the map of the application domain and following this only low voltage read operations are needed for various application cases such as drone localization and path planning.

[R31] *Sci. Adv.* 2021, 7, eabg9180

[R32] *Adv. Mater.* 2023, 35, 2207006

Comment #5) I recommend splitting the "Introduction" section into two distinct sections: a) Introduction, and b) Literature Review. This adjustment will facilitate reader comprehension.

Response:

We appreciate Reviewer #1's constructive and encouraging comments. We divide the introduction with two section, which reviewers suggested. Additionally, as the reviewer point out on comment#1, it further described the latest research on Gaussian transistors.

We revised the manuscript as follows:

Page 4, 1st paragraph,

Consequently, without innovative technologies that can dramatically reduce resource demands when dealing with high-dimensional complex distribution functions like GMMs, the advantages of probabilistic inference procedures remain out of reach for numerous applications.

Literature Review

Pursuing a probabilistic inference, significant strides have been made in recent ...

Page 4, 2nd paragraph,

For instance, analog circuits based on silicon-based complementary metal–oxide–semiconductor (CMOS) technology have been used to emulate Gaussian functions. However, in conventional Si metal-oxide-semiconductor field-effect transistor (MOSFET)-based circuits, at least four transistors are required to represent a Gaussian distribution^{29, 30}. Michail *et al.* presented a GMM-based classifier by connecting NMOS and PMOS bump circuits with a winner-takes-all (WTA) circuit.³⁶ Another study, Vassilis *et al.* proposed low-power bell-shaped analog classifiers (CLFs) by implementing GMM (machine learning) circuits with fewer number of transistors and three input values, and tested on real-world biomedical classification problems.³⁷ Nevertheless, to tune the characteristics of the distribution function, the number of

transistors increases often to more than 10, rendering the circuit quite complex and inevitably increasing power consumption^{26, 27, 31, 32}.

Comment #6) A thorough grammar and English check is necessary. For instance, in Figure 2, "stricture" is a different word than "structure." Please rectify this.

Response:

We appreciate Reviewer #1's attentive review. We also rechecked the whole manuscript thoroughly and corrected all errors and typos as much as we can.

We revised the figure as follows:

We revised the manuscript as follows:

Page 3, 2nd paragraph,

By combining multiple Gaussian functions through a mixture, multi-modal statistical data can be efficiently modeled. Nevertheless, evaluating functions of Gaussian mixture model (GMM) using a digital **data-path** often results in excessive computational workload.^{22, 23}

Page 7, 3rd paragraph

Over 100 consecutive operations, only marginal **changes** in μ (less than 100 mV) and σ (less than 30 mV) **was** observed. (Fig. **2f**) Furthermore, uniform operation was demonstrated across 8 devices, securing the potential for large-scale circuits. (Supplementary Fig. 7)

Page 8, 3rd paragraph,

When a programming voltage is applied, electrons (or holes) trapping occurs through tunneling, similar to conventional **floating gate** memory^{46, 48, 54}

Page 10, 1st paragraph,

As shown in Fig. **4e**, due to the V_T shift induced by the amount of trapped **charges**, a near-linear relationship between μ and the programming voltage was achieved.⁵⁴

Page 13, 1st paragraph,

Fig. **6a** only increases the number of GMT devices but still requires three logarithmic analog-to-digital converters (log-ADCs) and three digital-to-analog converters (DACs). Supplementary Table **1 and 2** capture the energy dissipation of digital **data-path** and proposed architecture for computing the likelihood from a 100-mixture component GMM.

For computing the likelihood, the digital **data-path** requires thousands of addition/subtraction and comparison, and hundreds of multiplications and read operations. Such extensive workload results into **~941 pJ** energy dissipation. Meanwhile, in Supplementary Table **2**, only a handful of operations are needed for the proposed GMT-based architecture since the Gaussian-like computations can be directly mapped to the characteristics of the device, only requiring **~18.33 pJ** energy. Thus, in Fig. **6i**, the proposed scheme requires about 118 times lower energy than the digital **data-path**.

Page 13, 3rd paragraph,

where I_0 is the peak current magnitude which is invariant to V_{μ} and only depends on the transistor size and V_D . σ^+ is a fitting parameter modeling I_D roll-off when $V_G > V_{\mu}$. σ^- models I_{DS} roll-off when $V_{GS} < V_{\mu}$.

Page 15, 3rd paragraph,

. To verify the validity of the travel path between two adjacent points, the sampled points were interpolated in 3D and applied to the GMT array, while the array's response output current was characterized.

Comment #7) As readers can observe, in the literature (reference 28), the characteristics of the Gaussian curve are independent. It is not clear from Figures 3 and 4 whether this also holds true for your work. Please provide clarification.

Response:

As the Reviewer #1 suggested, to ensure the clarity of the results, it is crucial to clarify the purpose of the study, which is to develop hardware that enables probabilistic inference operation using Bayesian filtering. For this purpose, we focus on adjusting mean (μ) and standard deviation (σ), which critically determine the Gaussian-like output shape for the desired inference operations. When regulating μ through the programming operation of the separate floating gates with the same polarity, σ variation was less than ± 0.05 V, which is practically negligible compared to the μ range adjustability (~ 4.3 V). On the other hand, μ remained nearly constant with the variation less than ± 0.07 V during the σ regulation procedure through the programming of the separate floating gates with the opposite polarity (σ adjustability of 0.3 V). These results confirmed the practically independent controllability of μ and σ in our GMT devices.

Moreover, our simulation results indicated that its controllability provides sufficient flexibility for the studied test-cases, resulting in a comparable accuracy to conventional GMM model-based predictions where μ and σ are independently programmed. Notably, our simulation model in Eq. (2) accounts for the dependence of μ and σ as

$$I_D = I_0 \exp\left(-abs\left(\frac{V_G - V_\mu}{\sigma^\pm(V_\mu)}\right)^\eta\right) \quad (2)$$

Here, σ^+ is a fitting parameter modeling I_D roll-off when $V_G > V_\mu$. σ^- models I_{DS} roll-off when $V_{GS} < V_\mu$. Notably, in our implementation, Expectation-Maximization (EM) algorithm used to train the studied application can recruit more elementary Gaussian-like functions, each mapped through a single GMT device, to compensate for the lack of flexibility in GMT than in a Gaussian function.

We revised the manuscript as follows:

Page 9, 2nd paragraph,

Beyond the accomplished control of Gaussian-like distribution through individual FG programming, the concurrent programming through the desirable ways allows for the meticulous manipulation of μ or σ , while ensuring minimal interference with other parameters (Fig. **4a,b**). For probabilistic inference operation applying Bayesian filtering, it is essential to adjust μ and σ in a way that can avoid interference between them. Therefore, the programming process is designed to minimize variations in σ when μ changes, and vice versa, ensuring that changes in σ result in minimal alterations to μ . Based on the programming method suggested above, we devised two distinctive programming methods to control the Gaussian-like distribution of our GMT device: i) Case 1: the injected charges have the same polarity and quantity, an equal amount of V_T shift arises in each channel layer ($\Delta V_{T,P} = \Delta V_{T,N}$).

Comment #8) Have you explored whether this method can be combined with a different ML model? I recommend consulting related literature. For example, please refer to the following work to understand the use of GMT in different classifiers: "General Methodology for the Design of Bell-Shaped Analog-Hardware Classifiers."

Response:

We agree with the reviewer's suggestion to explore connections between GMT and machine learning (ML) models to further substantiate the impact of proposed work.

Beyond the studied test-cases, we believe the proposed GMT devices would be able to enhance various other ML tasks disruptively. For example, several speech recognition models integrate GMMs with Hidden Markov Models (HMMs) for robustly handling audio feature variations ^[R33]. The proposed GMTs can efficiently process large scale GMMs to improve robustness and scalability of these speech recognition models at the edge. In natural language processing (NLP), Gaussian function-based techniques are used for tasks like word embedding and semantic analysis for deciphering language patterns ^[R34]. These tasks can be accelerated by GMTs.

Gaussian function's ability to model uncertainties and variations in sensor data as well as learning models is also instrumental in developing more robust, risk-aware algorithms. Notably, a variational inference of ML models has gained attention where Gaussian functions are used to approximate complex posterior distributions, allowing for more efficient and scalable Bayesian analysis in large data sets ^[R35]. By storing Gaussian function-based complex posterior distributions, GMT arrays would also be able to accelerate variational inference tasks through in-memory computing. GMTs can also be used to accelerate Gaussian processes (GPs) which are a powerful statistical method for tasks like regression, classification, and optimization ^[R36]. GPs allow for flexible, non-parametric modeling of data, providing not only predictions but also quantifying uncertainty in those predictions, which is particularly valuable in fields like robotics, time-series forecasting, and spatial data analysis.

We also provided additional details on the latest research regarding Gaussian function circuit with using various ML models to enhance the clarity of the discussion in the main text.

Page 4, 2nd paragraph,

For instance, analog circuits based on silicon-based complementary metal-oxide-semiconductor (CMOS) technology have been used to emulate Gaussian functions. However, in conventional Si metal-oxide-semiconductor field-effect transistor (MOSFET)-based circuits, at least four transistors are required to represent a Gaussian distribution^{29, 30}. Michail *et al.* presented a GMM-based classifier by connecting NMOS and PMOS bump circuits with a winner-takes-all (WTA) circuit.³⁶ Another study,

Vassilis *et al.* proposed low-power bell-shaped analog classifiers (CLFs) by implementing GMM (machine learning) circuits with fewer number of transistors and three input values, and tested on real-world biomedical classification problems.³⁷ Nevertheless, to tune the characteristics of the distribution function, the number of transistors increases often to more than 10, rendering the circuit quite complex and inevitably increasing power consumption^{26, 27, 31, 32}.

[R33] *IEEE Transactions on Speech and Audio processing*, 2004, 13, 1, 14-22.

[R34] *Guide to big data applications*, 2018, 83-104.

[R35] In International conference on machine learning, 2015, 1613-1622. PMLR.

[R36] *International journal of neural systems* 2004, 14, 02, 69-106.

Comment #9) Have you evaluated your GMT over corners or in the presence of PVT variations?

Response:

Thank you very much for the insightful comments from Reviewer #1. Unfortunately, electronic devices based on organic materials face challenges in keeping up with the performance of silicon-based electronic devices, not only in electrical performance but also in various aspects of the process. Consequently, there is a scarcity of cumulative plot studies in organic electronic devices that can be compared directly with silicon-based devices.

The measurement of PVT (process, voltage, and temperature) variations in organic electronics systems poses several challenges. Most studies have focused on measuring small quantities of process variation at the laboratory scale and have not investigated voltage or temperature variations in the circuit level.^[R37-41] Organic molecules generally exhibit lower thermal stability compared to inorganic materials. This characteristic makes it difficult to observe and quantify changes in the behavior of organic electronic systems with temperature variations, especially over a wide range. Moreover, to examine the operation of a circuit with multiple components, the components must be fabricated in a consistent manner, and the circuit

connecting them must also be constructed and measured. However, at the laboratory scale, measuring these variations is practically affordable or at least extremely challenging. Indeed, to the best of our knowledge, despite our diligent exploration, it was challenging to find a study that comprehensively investigates the PVT variation characteristics of organic devices. This could be due to limitations in the available measurement techniques, equipment, or the inherent complexity of dealing with organic materials. We hope the reviewer's kind understanding of our situation.

Due to the challenges in some aspects described above, we could not provide an accurate assessment of PVT variation in our organic electronic system. However, as our efforts to provide the best possible response to Reviewer #1's comment, we analyzed the device-to-device uniformity and reproducibility of the devices based on iCVD insulating layers. As shown in Fig R3 and Supplementary Fig 7, the fabricated devices including 16 metal-insulator-metal devices with p(CEA-co-DEGDVE) BDL and pV3D3 TDL, and 8 GMT devices exhibited highly uniform electrical characteristics, which is usually quite tricky to achieve in organic electronic devices. Through this, we hope Reviewer #1's kind consideration on the novelty and excellence of this study.

Fig. R3. a, C - V characteristics of 16 different Al/p(CEA-co-DEGDVE)(100nm)/Al devices and **b**, capacitance depend of number of MIM device. **c**, C - V characteristics of 16 different Al/pV3D3(14nm)/Al devices and **d**, capacitance depend of number of MIM device

Supplementary Fig. 7. a, Transfer characteristics with 8 different devices. **b**, Fitting result of mean (μ) and standard deviation (σ) values.

Meanwhile, we have accounted for the impact of process variability in GMT by considering perturbations in the modelled curves and analyzing their impact on the prediction accuracy. In Fig. R4(a), various parameters of the modeled curve (μ , σ^+ , σ^-) as well as curve height were randomly perturbed by following a Gaussian distribution as shown at the top of the figure. Resultant distribution of perturbed GMT curves is also shown in Fig. R4(a). Fig. R4(b) shows the prediction error for drone tracking (main manuscript, Fig. 6f) at increasing degree of perturbation by increasing the constant K of variations in Fig. R4(a). It is noteworthy, that even when K reaches to 20%, the prediction errors only gracefully degrade. Such resilience of probabilistic inference is attributed to their iterative perception-hypothesis loops where the impact of processing errors can be mitigated by a sequential decision making on the net outcome, unlike the typical feed forward deep learning models where sensing and processing errors can directly impact the outcome.

Fig. R4. a, Random perturbations considered on the modeled GMT curve to emulate the impact of process variability **b**, Prediction error in drone tracking at increasing perturbation constant.

[R37] *IEEE Journal on emerging and selected topics in circuits and systems*, 2017, 7 (1), 7-26.

[R38] *Org. Electron.*, 2014, 15, 701–710

[R39] *Org. Electron.*, 26 (2015) 371–379

[R40] *Adv. Mater.*, 2014, 26, 5722–5727

[R41] *Nano Lett.*, 2013, 13, 3864–3869

[R37] *IEEE Journal on emerging and selected topics in circuits and systems*, 2017, 7 (1), 7-26.

[R38] *Org. Electron.*, 2014, 15, 701–710

[R39] *Org. Electron.*, 26 (2015) 371–379

[R40] *Adv. Mater.*, 2014, 26, 5722–5727

[R41] *Nano Lett.*, 2013, 13, 3864–3869

Comment #10) Please include a comparison table with related works and corresponding metrics. This addition will enhance the overall quality of your work.

Response:

To emphasize the advantages enabled by the newly proposed device architecture, we added two tables, one comparing the performance with previously reported anti-ambipolar transistors, and the other comparing with CMOS devices capable of producing Gaussian-shaped output.

The GMT device developed in this study shows a relatively low operating voltage compared to the anti-ambipolar transistors utilizing various channel materials employed in the previous studies. In addition, in comparison to the previously reported CMOS-based Gaussian-shaped circuits, the GMT device exhibits a significant advantage in reducing the complexity of Gaussian circuits for stochastic inference due to its simple 3-terminal, single transistor structure.

We added the two tables in Supporting Information as follows:

Added the reference [18-39] in the revised Supporting Information.

Table S4. Summary of the electrical characteristics of the reported anti-ambipolar transistors

Year ^[ref]	Operating voltage	Semiconductor	Dielectric	Controllability
2016 ^[18]	40	MoS ₂ /WSe ₂	SiO ₂	X
2018 ^[19]	80	MoS ₂ /tetracene	SiO ₂	X
2018 ^[20]	10	α -6T/PTCDI-C8	PMMA/Al ₂ O ₃	Channel length
2019 ^[21]	50	PTCDI-C13/DNTT	SiO ₂	X
2019 ^[22]	20	MoS ₂ /BP	SiO ₂	Top gate
2020 ^[23]	60	H-Type/TTT-CN	SiO ₂	X
2020 ^[24]	6	MoS ₂ /CNT	Al ₂ O ₃	Top gate
2020 ^[25]	10	- (simulation)	PMMA/Al ₂ O ₃	X
2021 ^[26]	60	PTCDI-C13/DNTT	SiO ₂	X
2021 ^[27]	50	PTCDI-C13/DNTT	-	Channel length

2021 ^[28]	30	InSe/WSe ₂	hBN	Strain
2021 ^[29]	80	inSe/2H-MoTe ₂	SiO ₂	Light
2022 ^[30]	15	WSe ₂ /ReS ₂	SiO ₂	Top gate
2022 ^[31]	40-100	DBTTF/cyh-NDI	SiO ₂	X
This work	10	PTCDI-C13 /pentacene	iCVD dielectric	Programming

Table S5. Summary of the performance of reported Gaussian like *I-V* characteristics system

Ref num	Number of Transistor	Input variable	Regulate factor
1975 ^[32]	4	3	μ
1993 ^[33]	7	3	μ
2013 ^[34]	22	5	μ, σ, A
2014 ^[35]	31	8	μ, σ, A
2019 ^[36]	15	4	μ, σ, A
2021 ^[37]	14	5	μ, σ, A
2022 ^[38]	18	4	μ, σ, A
2023 ^[39]	10	3	μ, σ, A
This work	1	1	μ, σ, A

18. *ACS Appl Mater Interfaces* 2016, **8**(24): 15574-15581.

19. *ACS Appl Mater Interfaces* 2018, **10**(38): 32556-32566.

20. *Nano Lett* 2018, **18**(7): 4355-4359.

21. *Adv Mater* 2019, **31**(29): e1808265.

22. *Nat Commun* 2019, **10**(1): 4199.

23. *J Mater Chem C* 2020, **8**(13): 4303-4308.

24. *Nat Commun* 2020, **11**(1): 1565.

25. *Adv Electron Mater* 2020, **6**(3).
26. *Appl Surf Sci* 2021, **542**.
27. *IEEE Electron Device Lett* 2021, **42**(9): 1323-1326.
28. *ACS Nano* 2021, **15**(5): 8686-8693.
29. *J Mater Chem C* 2021, **9**(32): 10372-10380.
30. *Adv Electron Mater* 2022, **9**(1).
31. *Adv Electron Mater* 2022, **9**(1).
32. *Electronics letters* 1975, **1**(11): 14-16.
33. *Proceedings of International Joint Conference on Neural Networks*, 1993,1, 475-479.
34. *Neurocomputing* 2013, **118**: 329-333.
35. *Neurocomputing* 2014, **138**: 69-77.
36. *Analog Integrated Circuits and Signal Processing*, 2019, **102**(2), 323-330.
37. *2021 34th SBC/SBMicro/IEEE/ACM Symposium on Integrated Circuits and Systems Design (SBCCI)*, 2021, 1-6.
38. *2022 Panhellenic Conference on Electronics & Telecommunications (PACET). IEEE*, 2022, 1-4.
39. *Electronics*, 2023, 12, 4211

Comment #11) I suggest improving the text size in the main figures, as it appears too small in certain cases.

Response:

We appreciate Reviewer #1's attentive comment. We revise the text size of main figures.

We revised the figures as follows:

We also rechecked the whole figures (include supporting information) text size and corrected all errors and typos as much as we can.

Comment #12) Please refine the way you articulate your work. Utilize third-person language rather than first-person. For instance, instead of saying "We believe that we can offer a fundamental platform," explain why you can offer this fundamental platform and support it with your results.

Response:

We fully agree with the Reviewer #1's comment about what we did not exactly describe. As pointed out, we modified the whole manuscript from first-person language to third-person language. Furthermore, we revised the latter part of the Conclusion section to emphasize the advantages of GMT devices in providing Bayesian computational capabilities and successful implementation of Bayesian operations using actual data from GMT devices.

We revised the manuscript as follows:

Page 16, 2nd paragraph,

Moreover, the GMT devices maintain their performance even with 2.0% of the applied tensile strain. Using the GMT device array and by co-designing the probabilistic inference model to device characteristics, we showed applications towards 3D probabilistic localization and path finding for drones. The GMT device, featured by a simple unit transistor design with a heterojunction channel, is capable of representing output values in the form of a probability distribution function. Moreover, the separate floating-gate structure allows for the programmable modification of the probability distribution function without the need for additional components. Furthermore, the new reconfigurable device design in this study has a great potential for the application to various probabilistic inference computing fields with increased computational

efficiency and integration density thereof. The successful implementation of inference operations was indeed demonstrated by using actual data obtained from GMT devices. Therefore, the newly proposed heterojunction-based, Gaussian-like transistor embedding a separate floating gate memory structure could serve as a powerful platform for Bayesian operations. For instance, they can improve the robustness and scalability of speech recognition models that use GMMs and HMMs by efficiently processing large-scale GMMs. In natural language processing, GMTs can accelerate tasks such as word embedding and semantic analysis, thanks to Gaussian function-based techniques. Additionally, GMTs can expedite variational inference, Gaussian processes, and other algorithms by leveraging Gaussian functions to model uncertainties, making them valuable for Bayesian analysis, regression, classification, and optimization in fields like robotics, time-series forecasting, and spatial data analysis.

Reviewer (#2)'s COMMENTS:

Manuscript "Highly Parallel, Ultra-Low-Power Probabilistic Reasoning with Programmable Gaussian Memory Transistors" by Lee *et al.* describes an experimental realization of a simple transistor-like device that produces a Gaussian-like response with controllable mean and variance. Although there are other realizations of a Gaussian-like response in analog devices, but they require a fairly complicated circuit; the demonstrated device thus can greatly decrease the required power requirements. Utility of networks of such devices for 3-dimensional data analysis/learning tasks has also been demonstrated. This is an impressive achievement, and will be of interest to the broad readership of *Nat. Comm.* However, I would like the authors to clarify/address several issues before the manuscript is published.

Response:

We appreciate the Reviewer #2's encouraging and constructive comments. We present the point-by-point response for each comment as follows.

Comment #1) Experimental realization of GMT in this work used organic semiconductors. Then potential performance advantages were modeled assuming 45 nm CMOS realization of GMT. Better motivation and more details are needed to place such estimates in proper context. For example, why 45 nm were assumed? Is it because the floating gate transistors cannot be realized using current state-of-the-art feature sizes (< 10 nm)? In any case, the comparison of GMT array-based vs digital realization of Gaussian Mixture Model should use state-of-the-art for the latter (≤ 5 nm). Also, speed/latency comparison is unrealistic since all modern (at least, high-power) digital processors are data-parallel hence latency modeling needs to take that into account.

Response:

We appreciate Reviewer #2's valuable comments. We have conservatively projected the proposed Gaussian-like memory transistor (GMT)-based platform to 45 nm channel length technology since at this channel length, commercial products at 45 nm requiring a similar floating-gate structure, such as NAND/NOR Flash memories, already exist ^[R42]. We also fully agree with the Reviewer's comment that a fair comparison to digital design must be made at advanced complementary metal-oxide-semiconductor (CMOS) processes.

We have, therefore, revised our comparison table as below:

Table R1. Compare of energy consumption of GMT and CMOS devices

	GMT (45 nm)	Digital CMOS (14 nm)	Digital CMOS (7 nm)
Energy/ Likelihood Step	23.63 pJ	101.7 pJ	27.1 pJ
Improvement		4.3×	1.14×

Notably, even though, we have conservatively projected our GMT-based design to 45 nm, novel technologies for the scalability of floating gate and memory gate stacks are under development that will allow aggressive channel length scaling of our GMT, thereby significantly enhancing their energy efficiency. For example, charge trap transistors (CTT) were demonstrated in Khan *et al.* demonstrating high-*k* metal/gate stack programmability at 14 nm node.^[R43] NAND flash technology with programmable gate-stacks has also been scaled to less than 14 nm node in industry research.^[R44] Therefore, it is anticipated that our GMT device can be further improved in terms of energy consumption through scaling down the devices.

Furthermore, we also agree with the reviewer's comment on latency comparison. Our prior latency comparison was under the assumption that digital CMOS only uses a single core processing. This was to show a stark contrast between both designs where GMT array-based processing is naturally parallel on a crossbar, whereas for digital CMOS multiple identical copies of processing core must be implemented, thus incurring proportional area overhead. Due to the discrepancy in circuit designs, we chose not to include a direct latency comparison in the table as far as both designs are not physically implemented. We appreciate the understanding of Reviewer #2 in this issue.

We revised the figure as follows:

Figure 6i, Table of improvement in key performance metrics for likelihood computation using 100-mixture, 3-dimensional density function at 45 nm technology node.

We revised the manuscript as follows:

Page 11, 2nd paragraph,

This novel method leads to a remarkable reduction of **over the 50 times** in power consumption and latency than other conventional CMOS techniques

Page 13, 1st paragraph,

For computing the likelihood, the digital **data-path** requires thousands of addition/subtraction and comparison, and hundreds of multiplications and read operations. Such extensive workload results into **~941 pJ** energy dissipation. Meanwhile, in Supplementary Table 2, only a handful of operations are needed for the proposed GMT-based architecture since the **Gaussian-like** computations can be directly mapped to the characteristics of the device, only requiring **~18.33 pJ** energy. Thus, in Fig. 6i, the proposed scheme requires about **50 times** lower energy than the digital **data-path**. Furthermore, when compared to the state-of-the-art 14nm and 7nm Digital CMOS, it also exhibited slightly improved performance (Supplementary Table 3).

We revised the supporting information as follows:

Page 24, 1st paragraph,

We have carefully conceived a platform based on Gaussian-like memory transistors (GMT) for implementation in technology with a channel length of 45 nm.

We have carefully conceived the envisioned Gaussian-like memory transistor (GMT)-based platform for implementation in the 45 nm channel length technology. This choice is grounded in the fact that commercial products utilizing a comparable floating gate structure, such as NAND/NOR Flash memories, are already present at the 45 nm channel length.

Accordingly, power dissipation was estimated using SPICE simulations, relying on predictive technology models referenced in FreePDK^{45TM} 45nm

(<https://eda.ncsu.edu/freepdk/freepdk45/>). **Additionally, for a fair benchmark against digital**

design, we referred to advanced studies. Khan *et al.* demonstrated high-k metal/gate stack programmability at the 14 nm node,¹⁶ and industry research presented NAND flash technology scaled to less than the 14 nm node.¹⁷ Therefore, we conducted another digital design of state-of-the-art complementary metal–oxide–semiconductor (CMOS) processes at both 14 nm and 7 nm. The GMT device still exhibited enhanced performance compared to digital CMOS devices under 14 nm, as illustrated in Table S3.

We revise and added the Compare of energy consumption of GMT and CMOS devices (Table R1) to Supplementary Table 1,2 and 3 in Supplementary Information (p. 26,27)

Table S1: 8-bit Digital Datapath’s Energy for Likelihood Computation using 100-Mixture, 3-Dimensional Gaussian Mixture Model (GMM) in CMOS 14 nm & CMOS 7 nm.

Operation Type	# of Operations	Energy/Op. (fJ) (CMOS 7 nm)	Energy/Op. (fJ) (CMOS 14 nm)	Energy/Op. (fJ) (CMOS 45 nm)
ADD/SUB	10699	0.325 ^{a)}	0.72 ^{a)}	30 ^{c)}
Multiplication	600	1.004 ^{a)}	3.307 ^{a)}	200 ^{c)}
Read	100	230 ^{b)}	920 ^{b)}	5000 ^{d)}
Total Energy		27.08 pJ	101.68 pJ	941 pJ

Table S2: 8-bit GMT-based Mixed-Signal Datapath’s Energy for Likelihood Computation using 100-Mixture, 3-Dimensional Density Function in Eq. 3 at 45 nm.

Operation	Energy/Operation (pJ) ^[e,f]	# of Operations	Energy (pJ)
8-bit DAC	0.94 ^{e)}	3	2.82
Voltage Gain	1.98	3	5.94
GMT Array Dynamic	2.15	1	2.15
GMT Array Bias	5.04	1	5.04
8-bit log-ADC	2.38 ^{f)}	1	2.38
Total Energy			18.33 J

- a) *IEEE Transactions on Circuits and Systems II: Express Briefs* 2015, **62**(8): 761-765.
- b) https://engineering.virginia.edu/sites/default/files/common/VLSI_2018.pdf
- c) *Electronics* 9, no. 1 (2020): 134.
- d) <https://www.cl.cam.ac.uk/teaching/1213/SysOnChip/materials/sg7power/zhp802be4b0.html>
- e) *IEEE Journal of Solid-State Circuits* 2014, **50**(2): 543-555.
- f) *IEEE Journal Of Solid-State Circuits* 2009, **44**(10): 2755-2765.

Table S3. Compare of energy consumption of GMT and CMOS devices

	GMT (45 nm)	Digital CMOS (14 nm)	Digital CMOS (7 nm)
Energy/ Likelihood Step	18.33 pJ	101.7 pJ	27.1 pJ
Improvement		5.5×	1.48×

[R42] *IEEE Journal of Solid-State Circuits*, 2008, 44, 1, 208-216.

[R43] *IEEE Electron Device Letters*, 2016, 38, 1, 44-47.

[R44] *IEEE Solid-State Circuits Magazine*, 2022, 14, 2 ,56-65.

Comment #2) The manuscript uses mixture of Gaussian vs Gaussian-like to characterize the device. The output is clearly not Gaussian. Even though in the analog community such response is modeled by Gaussians, that's a crude approximation; it might be OK for extremely crude tasks where only a few bits of information are available, but for other tasks of interest to the readership of Nat. Comm. (e.g., physical simulation) the device response will need to be modeled more precisely. Intuitively it seems that the Gaussian-like response is obtain via competition of 2 channels so it should be modeled well as a product of 2 Fermi-Dirac distributions with opposite signs of variables and 2 different "variances". Indeed, playing around with such distribution seems to produce shapes that look more like the asymmetric responses measured here for GMT than the produced Gaussian fits. The tails of the distributions are clearly *exponential* (i.e., $\exp(-k x)$) not Gaussian $\exp(-k x^2)$. So clearly describing the device response as "Gaussian" is a stretch, and the device should be termed "Gaussian-like" Memory Transistor. Consequently, I suggest that the authors revisit the awkward design of the function in Eq. (2); at the very least there needs to be a comparison of use of Eq. (2) vs. regular Gaussian. P.S. Of course, for the crude learning tasks the differences do not matter, but this points out that using Gaussians might be suboptimal for the implementation on a digital device and functions that are more economically implementable can be used. The authors might want to comment on this.

Response:

We fully agree with the Reviewer #2 that the response from GMT device is only "Gaussian-like" and not a true "Gaussian" function. Specifically, the proposed device produces heavier tails than a true Gaussian function. We acknowledge the Reviewer #2's comment and therefore have been careful to refer to the device response as "Gaussian-like", rather than "Gaussian", in the manuscript. We have now more strongly emphasized this critical discernment. The Eq. (2) also accounts for this discrepancy between true Gaussian and GMT's Gaussian-like behavior:

$$I_D = I_0 \exp\left(-abs\left(\frac{V_G - V_\mu}{\sigma^\pm(V_\mu)}\right)^\eta\right) \quad (2)$$

Notably, in the above equation η is a fitting parameter, modeling the power law index of $\log-I_D$ and V_G dependence, and has a value of 1.2. This indicates that the true response of the device lies in between Gaussian and exponential tails. Notably, GMT encapsulates two transistors within one structure. Therefore, the device's response leans more towards exponential tails rather than being exponential itself (i.e., $\eta=1$) as would be expected in a device that completely follows Fermi-Dirac distribution-induced carrier injection to the channel. [R45-46]

Please note that our comparisons on drone position tracking and path planning already account for the Reviewer's suggestion of comparing the algorithmic performance of our device's Gaussian-like response against predictions from a functionally precise Gaussian mixture model. A comparable performance is obtained from both models due to their qualitative similarity. Notably, the current response of GMT is also single mode and decays exponentially – similar to a Gaussian function. The expectation-maximization (EM) algorithm adopted to train the models based on GMT's Gaussian-like response is thus able to define GMT's mean aligning with the centroid of a cluster of point-cloud modelled by a GMT. Adding on to the Reviewer#2's remark, while a Gaussian function has analytically defined statistical properties, such as its higher-order moments and well-defined conjugate functions, the response of GMT does not lend itself to such a nice analytical expression. Therefore, while computational efficiency of training with Gaussian function-based models is high, in our case numerical extraction of properties of interest such as likelihood is necessary. This results in a higher training cost of models realized with GMT's response. Nevertheless, since for a vast majority of applications, inference is the dominant computation and training can be performed on cloud, we believe that even with its *Gaussian-like response*, GMT can significantly impact these application spaces.

We revised the manuscript as follows:

Title,

Highly Parallel, Ultra-Low-Power Probabilistic Reasoning with Programmable Gaussian-like Memory Transistors

Abstract,

line 6,7 → a Gaussian-like memory transistor (GMT) is proposed ... where a programmable Gaussian-like current-voltage response

line 9 → which allows for the exquisite control of the Gaussian-like current output

line 11 → The GMT could effectively modulate the Gaussian-like distributions ...

line 19 → avoidance tasks is demonstrated using Gaussian-like curves from GMT

line 20 → programmable Gaussian-like outputs, the 3-terminal GMT device developed

Page 5, 2nd paragraph,

This article proposes a new type of device, called Gaussian-like memory transistor (GMT), Moreover, each semiconductor has a separate floating gate (FG), allowing independent adjustment of their channel conductivity, thereby enabling systematic control of Gaussian-like expression through the FG. ... The GMT devices possess the remarkable capacity to adjust Gaussian-like shapes extensively, all the while maintaining an impressively low power consumption profile

Page 6, 2nd paragraph,

For systematic controllability of Gaussian-like output characteristics by an independent modulation of p- and n-type channel conductance, separate FG structures were introduced beneath both p- and n-type semiconductors

Page 9, 2nd paragraph,

Beyond the accomplished control of Gaussian-like distribution through individual FG programming, the concurrent programming through the desirable ways allows for the meticulous manipulation of μ or σ , while ensuring minimal interference with other parameters (Fig. 4a,b). Based on the programming method suggested above, we devised two distinctive programming methods to control the Gaussian-like distribution of our GMT device: i) Case 1: the injected

charges have the same polarity and quantity, an equal amount of V_T shift arises in each channel layer ($\Delta V_{T,P} = \Delta V_{T,N}$).

Page 10, 1st paragraph

resulting in a parallel movement of the **Gaussian-like** curve towards the negative direction (Fig. **4c**). Likewise, when electrons were trapped in both FGs, the V_T shifted towards the positive V_G direction for both channel layers, leading to a parallel movement of the **Gaussian-like** curve towards the positive direction without changing the **Gaussian-like** curve shape (Fig. **4d**).

Page 13, 1st & 3rd paragraph

Meanwhile, in Supplementary Table **2**, only a handful of operations are needed for the proposed GMT-based architecture since the **Gaussian-like** computations can be directly mapped to the characteristics of the device, only requiring ~ 18.33 pJ energy.

At varying V_μ , $I_D - V_G$ characteristics resemble a skewed **Gaussian-like** function where the skewness itself is controlled by the peak gate voltage.

Page 14, 2nd paragraph

In Fig. 6, even though each GMT only emulates one-dimensional **(1D) Gaussian-like** function, ... Since the gate-controlled conductance of each GMT follows **a Gaussian-like** function.

Page 15, 2nd paragraph

Significantly, our GMT-based co-designed approach fulfils a comparable performance despite various non-idealities such as skewed mapping of a **Gaussian-like** function and limited programmability of only mean variables of the characteristics.

We also rechecked the whole caption of figures (include supporting information) and corrected all expression.

[R45] *IEEE Transactions on Electron Devices*, 1997, 44 (1), 89-96.

[R46] *Solid-State Electronics*, 2000, 44 (9), 1707-1710.

Comment #3) Equation (3) is extremely confusing to a non-specialist. Single product of sums of 1-d Gaussians can only model signal in a small domain such that the signal is very smooth. In a larger 3-d domain the signal must be approximated as a linear combination of 3-d Gaussians. Clearly, this is what is done in practice here, as Figure 6h suggests; this needs to be clarified.

Response:

We appreciate the Reviewer#2's comments and thank you very much for providing constructive suggestions regarding the lack of clarity in the equation. Our previous GMT-based map models were under a mean-field approximation where a multi-dimensional function $f(x,y,z)$ is approximated as $f(x,y,z) \sim f_x(x) \times f_y(y) \times f_z(z)$, i.e., each dimension of the function is treated independently. Although, this enabled simplicity of the implementation, as noted in our GMT architecture, the approximation can also introduce significant errors for certain mapping functions. For example, it leads to spurious peaks such as when mapping a function with peaks at $x, y = (1, -1)$ and $(-1, 1)$, approximated function has peaks at $x, y = (-1, 1)$ and $(1, -1)$ as well.

We have therefore now revised our previous proposition. In Fig. S19(a), consider a series connection of GMTs. Here, gate-controlled conductance of each GMT follows a Gaussian-like behavior as described in Eq. (2). At lower drain biasing voltages (V_{DS}), GMTs operate under diffusion and linearity of their current against V_{DS} improves as shown in Fig. S19(b). Therefore, for a series connection of GMTs, under lower column biasing voltage, the column current follows a *Harmonic mean of their Gaussian-like* (HMGL) characteristics as shown in Fig. S19(a). Fig. S19(c) and (d) show the surface and contour profile of the resultant HMGL function emulated by the column current of series-connected GMTs. Notably, similar to a Gaussian function, HMGL is a single mode function. However, unlike a Gaussian function, it follows rectilinear surface tails as shown in Fig. S19(d).

In Fig. 6(a) in the main manuscript, we utilize series-connected GMTs as a kernel function of the mixture model. In the shown architecture, the current of each column follows a 3-dimensional HMGL function. Column currents are summed by a downstream current buffer to emulate a mixture of HMGL functions. This mixture of HMGL functions is trained against the point cloud map of flying domain using Expectation-Maximization (EM) algorithm and the corresponding programming parameters of GMTs (i.e., μ) are learned. Using such map models based on a mixture of HMGL functions, in Figs. 6(a, b) we demonstrate the microarchitectures

of GMTs for drone tracking by particle filtering and path planning. In Fig. 6(f), map models based on a mixture of HMGL functions demonstrate a comparable performance to the traditional GMM-based models. Meanwhile, a critical advantage of the proposed architecture compared to a digital GMM data path in Fig. 6(c) is that all mixture functions can be processed simultaneously and only $N \times M$ GMTs are required for a N -dimensional mixture function with M components. Therefore, even very complex statistical mixture function models can be efficiently implemented under stringent area/power constraints.

We revised the manuscript as follows:

Page 14, 2nd paragraph

In Fig. 6, even though each GMT only emulates one-dimensional (1D) Gaussian-like function, the higher-dimensional maps can be efficiently implemented through an array of GMT devices, without requiring complex analog multiplications, by co-designing the map function to GMT characteristics. In Fig. 6a, consider a series connection of GMTs. Since the gate-controlled conductance of each GMT follows a Gaussian-like function, resultantly, their column current follows a harmonic mean of Gaussian-like (HMGL) function. HMGL is a multidimensional function, matching the dimension of map model, and is given by

$$\text{HMGL}(V_X, V_X, V_X) = \frac{1}{\frac{1}{G(\mu_X, \sigma_X, V_X)} + \frac{1}{G(\mu_Y, \sigma_Y, V_Y)} + \frac{1}{G(\mu_Z, \sigma_Z, V_Z)}} \quad (3)$$

Here, for Gaussian like function along the x -axis, μ_x and σ_x are the programmed parameters of GMT that are learned using expectation-maximization procedure.⁷⁵ V_X is the applied gate voltage. Likewise, Gaussian-like functions along y -axis and z -axis are defined. Utilizing a mixture of 3D HMGL functions, the 3D map of flying domain is modelled as

$$M(x, y, z) = \sum_{i=1}^N \frac{1}{\frac{1}{G(\mu_{Xi}, \sigma_{Xi}, V_X)} + \frac{1}{G(\mu_{Yi}, \sigma_{Yi}, V_Y)} + \frac{1}{G(\mu_{Zi}, \sigma_{Zi}, V_Z)}} \quad (4)$$

With sufficient number of HMGL functions, arbitrary complex maps can be modelled.

To compute the log-likelihood of a predictive hypothesis using the mapping model in Eq. (4), the pixel outputs of depth measurements are projected to 3D based on a position and orientation (i.e., pose) hypothesis for the localizing drone. The transformed outputs are converted to analog domain and applied to the crossbar using digital-to-analog converters (DAC). The

output current induced by the crossbar follows the log-likelihood of pose hypothesis as defined by Eq. (4).

We add figures in the supporting information as follows:

Supplementary Fig. 19. a, Series connection of GMTs, while biasing the column at low voltages, induces a Harmonic mean of GMT characteristics. **b**, The transfer curves of low drain voltage (V_D) 2.6V to 3V with the subthreshold operation. **c**, Surface plot and **d**, contour plot of normalized column current of three series-connected GMTs. The current profile shows a single mode peak and rectilinear surface tails.

REVIEWERS' COMMENTS

Reviewer #1 (Remarks to the Author):

Dear Authors,

I read both the manuscript and your comments.
I am very happy that you address all my concerns.
The text is well-written.
This paper can be accepted for publication

Reviewer #2 (Remarks to the Author):

I am pleased with the detailed response to my and other referee's comments on the original manuscript. I recommend acceptance of the manuscript for publication.